# The past, present, and future viscous heat dissipation available for Greenland subglacial conduit formation

Kenneth D. Mankoff[1] and Slawek M. Tulaczyk[2]

[1]Department of Geosciences, Pennsylvania State University, University Park, PA 16802
[2]Earth and Planetary Sciences Department, University of California Santa Cruz, Santa Cruz, CA 95064

*Correspondence to:* Ken Mankoff (mankoff@psu.edu)

**Abstract.** Basal hydrology of the Greenland Ice Sheet (GIS) influences its dynamics and mass balance through basal lubrication and ice/bed de-coupling, or efficient water removal and ice/bed coupling. Variations in subglacial water pressure through the seasonal evolution of the subglacial hydrological system help control ice velocity. Near the ice sheet margin, large basal conduits are melted by the viscous heat dissipation (VHD) from surface runoff routed to the bed. These conduits may lead to efficient drainage systems that lower subglacial water pressure, increase basal effective stress, and reduce ice velocity. In this study we quantify the energy available for VHD historically, at present, and under future climate scenarios. At present, 345 km$^3$ of annual runoff delivers 66 GW to the base of the ice sheet per year. These values are already ~50% more than the historical 1960 – 1999 value of 46 GW. By 2100 under IPCC AR5 RCP8.5 (RCP4.5) scenarios, 1278 (524) km$^3$ of runoff may deliver 310 (110) GW to the ice sheet base. Hence, the ice sheet may experience a 5-to-7-fold increase in VHD in the near future which will enhance opening of subglacial conduits near the margin and will warm basal ice in the interior. The other significant basal heat source is geothermal heat flux (GHF) which has an estimated value of 36 GW within the present-day VHD area. With increasing surface meltwater penetration to the bed the basal heat budget in the active basal hydrology zone of the GIS will be increasingly dominated by VHD and relatively less sensitive to GHF, which may result in spatial changes in the ice flow field and in its seasonal variability.

## 1  Introduction

Numerical models and observations of the Greenland Ice Sheet (GIS) link surface meltwater penetration to the bed to both short (hourly, daily) and long (seasonal, decadal) temporal variations in ice velocity (Zwally et al., 2002; Bartholomew et al., 2011; Banwell et al., 2013; Shannon et al., 2013; Mayaud et al., 2014; Tedstone et al., 2015). However, the link between increased basal water inputs and ice sliding is a complex one, largely because viscous heat dissipation (VHD) from water flow beneath ice may melt out efficient drainage tunnels whose presence may decrease, or even reverse, the tendency for ice flow to accelerate with increasing meltwater inputs to the bed (Kamb, 1987; Sundal et al., 2011; Tedstone et al., 2015).

Early in the melt season water is added to the subglacial system but cannot be efficiently removed, increasing subglacial water pressures and ice velocities. Later in the melt season, increased runoff causes efficient drainage conduits to form, at least near the ice sheet margin. These large drainage conduits reduce the subglacial water pressure and ice velocity (Hewitt, 2013).

Even efficient drainage conduits can at times become over-pressured, with associated increase in ice velocity, until basal water is removed or the conduit opens more from additional melting (Schoof, 2010). This dynamic subglacial hydrologic system influences ice velocity during winter months as well. This behavior is less well understood, but in years with above average summer acceleration there is evidence of below average winter velocities and a reduced net annual displacement (Sundal et al., 2011). In addition, a recent observational study shows a regional and decadal velocity decrease coincident with a 50% runoff increase in southwest Greenland (Tedstone et al., 2015). Farther from the margin there is less surface runoff and therefore less subglacial water available. In this interior region channelized flow and subglacial conduits may not form. If they do form, they will in general creep closed more quickly than conduits that form under the thinner ice near the margin. Less efficient drainage in the GIS interior, relative to the marginal zone, is one likely cause for the ice accelerations observed there under increased water inputs (e.g. Bartholomew et al. (2011); Shannon et al. (2013); Doyle et al. (2014)).

Studies examining surface melt, supra-glacial routing, subglacial hydrology, and the response of ice sheet outlet glaciers to those various inputs take place predominantly in southwest Greenland, focusing largely on the Russell, Leverett, Paakitsoq, or nearby glaciers (for example, Banwell et al. (2013); Arnold et al. (2014); Andrews et al. (2014); Tedstone et al. (2015)). Furthermore, present day weather, runoff, outflow, and other data are often used in those studies, since daily, hourly, or higher temporal resolution of the data is beneficial to the models. However, this approach limits the focus of these studies to recent seasons for which abundant in-situ sensor data exist. In order to examine future scenarios, Mayaud et al. (2014) built on the work of Banwell et al. (2013), but used a conduit model that includes melt opening and creep closure, driven by a positive degree day runoff model, to examine future changes to year 2095 under various IPCC RCP scenarios (Moss et al., 2010). Those models had hourly or daily resolution and were again limited to the southwest sector of the ice sheet.

Here we perform a broader analysis that uses runoff over the entire GIS on annual and decade timescales, and discuss changes between a historical baseline and the present, and between the present and future RCP 4.5 and 85 scenarios. We employ flow routing to distribute the surface meltwater under the GIS following the common assumption of subglacial water pressure distribution being determined by local ice weight, and track the available energy at the bed as water runs down the hydropotential, accounting for the water and ice phase transition temperature (PTT) variations due to changes in pressure. We frame the discussion in terms of changes in available power (Watts), rather than focusing on water pressure in a conduit relative to overhead ice pressure. We report both the total GIS-wide energy budgets, and its distribution per basin and at a 5x5 km grid resolution. We also highlight a high-resolution (150 m) calculation near Petermann Glacier and along a single subglacial water flow line in southwest Greenland.

## 2  Data

We use 150 m resolution basal topography and surface topography (IceBridge BedMachine Greenland, Version 2) from Morlighem et al. (2014, 2015) to calculate both surface and subglacial flow routing and subglacial pressures. Surface runoff, equal to the surface meltwater plus rain less refrozen water, comes from MAR v3.5.2 (Fettweis et al., 2013).

We report results for a historical period (1960 – 1999), the present (2010 – 2019), and IPCC AR5 RCPs 4.5, and 8.5 (2090 – 2099) (Moss et al., 2010). We also highlight a baseline (TB, 1985–1994) and reference (TR, 2007–2014) period that match the baseline and reference periods in Tedstone et al. (2015).

We process the entire GIS at 5 km resolution, the area near Petermann Glacier at 150 m resolution, and part of West Greenland (near the Russell and Leverett glaciers) and at 150 m resolution, where we extract the sample flowline segment.

## 3 Methods

### 3.1 Model Description

We use a flow-routing and energy balance model that incorporates common assumptions about glacier hydrology (e.g. Röth-lisberger (1972), Shreve (1972)), but does not explicitly resolve subglacial conduits. We lay out our methods and assumptions by tracing the path of a unit parcel of surface meltwater from source to sink (ice surface elevation at the origin of a meltwater parcel to submarine or terrestrial outlet where it discharges from the ice sheet).

In most cases we assume all surface runoff that begins at elevations above 2000 m is unable to leave the surface or penetrate to the bed at those elevations (Poinar et al., 2015). In these cases we route it on the surface to the 2000 m elevation contour. We also show the results for surface runoff not routed to the 2000 m contour, but only for the RCP 8.5 scenario, where runoff occurs at the highest elevations. Unless routed to the 2000 m contour, surface runoff is assumed to access the bed within the 5 km square grid cell in which it originates (Yang and Smith, 2013). In reality water may flow slightly farther before leaving the surface (Yang et al., 2015), but we ignore this horizontal transport because when streams do travel this far on the surface, they are most likely to do so in gently-sloped and crevasse-free regions (Poinar et al., 2015). Horizontal transport of surface meltwater with a small surface elevation drop implies only a small impact on the calculated VHD due to the ice surface, and may actually increase VHD if the ice thickens due to basal topography. Horizontal surface transport in an area with large surface slopes is unlikely, because such regions have high driving stress, and crevasses routing the stream to the bed are likely to be present.

Water storage may occur in firn or in crevasses. We assume these volumes are insignificant (at most a few percent) relative to the total runoff amount, that their relative importance is not likely to change much in the future, and that englacial storage does not release heat at the bed, which is the focus of this study.

Once at the bed, flow routing moves water down the gradient of the hydropotential $\phi$ (Shreve, 1972),

$$\nabla\phi = \nabla\phi_z + \nabla\phi_p = \rho_w\,g\,\nabla z_b + \alpha\,\rho_i\,g\,(\nabla z_s - \nabla z_b), \tag{1}$$

with $\phi$ the hydropotential (units Pa), $\phi_z$ the elevation component of the hydropotential, $\phi_p$ the pressure component, $\rho_w$ the density of meltwater (1000 kg m$^{-3}$), $g$ gravity, $z_b$ the bed elevation, $\alpha$ a flotation fraction (set to 0.9) because the subglacial system is often slightly less than the ice-overburden pressure (Engelhardt and Kamb (1997); Fountain (1994); Meierbachtol et al. (2013)), $\rho_i$ the density of ice (917 kg m$^{-3}$), and $z_s$ the surface elevation. All water is assumed to move to the one

neighboring cell with the lowest hydropotential with eight total neighbors considered. Flow routing is implemented using the `r.watershed` tool in GRASS GIS (Neteler et al., 2012) version 7.0.5, with $\phi$ as the "elevation" input with all local minima filled so that all water leaves the ice sheet (see Appendix A).

The total hydropotential can be decomposed into an elevation term, $\phi_z$ and a pressure term, $\phi_p$, where the former is the first term on the right hand side of Eq. 1 and the latter is the second term. We use this decomposition to examine in more detail the spatial distribution of flow driven by changes in the bed elevation, and flow driven by the pressure gradient.

The water remains at the pressure-dependent PTT and energy is released based on the change in hydropotential combined with the changing PTT. Because our focus is on energy available at the glacier bed, we ignore heat released due to changing PTT down the moulin (e.g. Catania and Neumann (2010)), and the model is initialized at the moulin bottom with a depressed PTT.

Our energy budget model tracks energy between inputs at the ice sheet bed where energy begins as either pressure or gravitational potential energy (which may be net positive if the source bed elevation is above the discharge elevation, or negative if it is below), and the output where the energy is in one of three forms: 1) the latent heat of cumulative basal melt caused by VHD released along the subglacial water flow pathway, 2) gravitational potential energy of discharge from land-terminating glaciers with terminii above seal level, or 3) pressure if discharged below sea level from a marine terminating glacier.

Between the input and discharge locations, all energy is assumed to dissipate at the base as heat within the grid cell where the energy transfer occurs (Isenko et al., 2005). As water flows down the hydropotential gradient, we track the energy released as heat, $Q$, based on the volume of water, the change in the hydropotential, and the change in the PTT,

$$Q = V(\nabla\phi - C_T c_p \nabla\phi_p \rho_w) \tag{2}$$

where $V$ is the volume of water (Fig. 1), $\nabla\phi$ the hydropotential gradient (Eq. 1), $C_T$ the Clausius-Clapeyron slope ($8.6\mathrm{x}10^{-8}$ K Pa$^{-1}$, Hooke (2005)), $c_p$ the specific heat of water (4184 J K$^{-1}$ kg$^{-1}$), $\nabla\phi_p$ the pressure component of the hydropotential gradient, and $\rho_w$ the density of water. The last term of Eq. (2) is the adjustment for the PTT, which increases the heat released along a flowline when the ice thickens downstream and the PTT drops, and decreases the heat released along a flowline when the ice thins downstream. If the second term on the right hand side of Eq. 2 is larger than the first term, then $Q$ is negative, and basal freeze-on occurs (Alley et al., 1998; Bell et al., 2014)

This model is driven by surface runoff, but includes flow routing at the bed, heat released due to either a gravitational potential energy drop or a pressure drop, and tracks the spatially varying PTT. However, this model does not directly represent conduits or track changing basal water pressures. This is because the results are presented on decade average timescales, while conduits and water pressures vary on hourly to seasonal timescales.

## 3.2 Comparison between model results and observations

We compare our results to those of Tedstone et al. (2015). We do this because the comparison provides a connection between the model results presented here, which are necessarily speculative due to limited observations outside the southwest sector and in the future, and independent observation-based results.

As mentioned in Sec. 2, we produce a Tedstone baseline (TB, 1985–1994) and a Tedstone reference (TR, 2007–2014) data set. Because Tedstone et al. (2015) reports a 50% increase in runoff over the 1985 – 2015 time period, we divide TR by TB to compare our VHD percentage increase or decrease to the Tedstone et al. (2015) increase. After comparing our model results against Tedstone et al. (2015), we then compute absolute difference (TR - TB), along with differences between our historical and present (P - H), and future and present (45 - P, 85 - P) periods.

It is important to note that while TB may match the baseline data from Tedstone et al. (2015), TR may not match the reference data from Tedstone et al. (2015). This is because the TB period is from ERA-Interim reanalysis products, but the TR period is from a "future" projection simulation (Fettweis et al., 2013), and is unlikely to have simulated the specific annual runoff, including the extreme melt in 2012 (Nghiem et al., 2012).

## 4 Results

### 4.1 Subglacial water volume

Annual average runoff volume has historically been 244 km$^3$, presently is 345 km$^3$, and in the future will be 524 or 1278 km$^3$ under the 4.5 or 8.5 scenarios. The spatial distribution of flux at the bed matches the large-scale surface distribution - more occurs in the south than the north, and the bulk occurs in the southwest sector. Most runoff also occurs at the edge of the ice sheet. Under RCP 8.5 it is predicted to occur at all elevations in south Greenland (Fig 2), but here we only show VHD results graphically for the scenario where water is routed on the surface to <2000 m elevation. Results are reported for water accessing the bed at all elevations in Table 1, column $8.5_{>2000}$.

Flow routing of basal hydrology causes orders of magnitude difference in water volume over small spatial distances as streams collect and discharge the water. At present, the largest volume of discharge is ~7 km$^3$ a$^{-1}$ from a single grid cell (2% of the total annual runoff). That percentage has not and does not significantly change under different RCP scenarios. The volume flow rate has increased from ~5 km$^3$ a$^{-1}$ historically, and is 10 and 31 km$^3$ a$^{-1}$ under the future 4.5 and 8.5 scenarios, respectively. The volumes do not change if flow begins at elevations >2000 m.

### 4.2 Pressure v. elevation driven flow

The zone around Greenland with active subglacial flow has distinct regions where the flow is driven by changes in elevation (Fig. 2 panel $\nabla \phi_z$) or pressure (Fig. 2 panel $\nabla \phi_p$). Flow leaves the ice sheet at the margin due to pressure-driven flow (i.e. from regions where ice thickness decreases downstream, red outer band in Fig. 2, panel $\nabla \phi_p$), but inland drainage can occur under regions where ice thickness and therefore pressure increases along a flow line (blue regions in Fig. 2). Distinct regions

of flow under thickening ice occur near the Petermann, Zachariae Isstrøm and 79 North Glaciers, and some coherent patches along the west coast. If water is allowed to penetrate to the bed from surface elevations >2000 m, then the southern sector has basal water flowing down-gradient but under thickening ice (Fig. 2). This mode of flow occurs predominantly in the interior regions, because the ice surface is flatter, the pressure gradient is reduced, and basal topography exerts a stronger control on the routing of the water.

Large differences in released heat (>35%) are due to flow under thinning or thickening ice. When water flows under thinning ice, ~35% of the heat released by the reduction in pressure is used to warm the water with the rising PTT (Röthlisberger, 1972), and subtracted from the VHD value (last term of Eq. (2)). In the regions highlighted above where flow occurs under thickening ice, a decrease in the PTT increases the VHD term.

## 4.3   Flow-routed spatial distribution of VHD

A spatial map of basal VHD is shown in Fig. 3 with the energy calculated based on Eq. (2). Summing the spatial data in Fig. 3 gives annual GIS-wide VHD of 46, 66, 110, and 310 GW for the historical, present, RCP4.5 and RCP8.5 cases respectively (Table 1). If runoff is not routed on the surface to the 2000 m elevation contour, there is no significant change. For the RCP 8.5 scenario, the total VHD only increases from 310 GW to 324 GW. There is a 5-fold increase in VHD between the present and the end of the century under RCP 8.5, or a 7-fold increase from the recent past.

At present a maximum up to ~1 W m$^{-2}$ is released where the largest volumes of water leave the ice sheet, over an entire 5x5 km grid cell. More generally, between 0.1 and 1 W m$^{-2}$ is released in the marginal zone, but by year 2100 under RCP8.5, this amount of heat is likely to be released throughout almost the entire area of GIS where runoff is projected to reach the bed. In the future, regions with high discharge may experience 10 W m$^{-2}$ of VHD.

In some regions, heating is "negative" which indicates basal freeze-on. These regions are a subset of the regions where flow is uphill (blue $\nabla\phi_z$ in Fig. 2) and pressure decreases due to ice thinning along-flow (red $\nabla\phi_p$ in Fig. 2). Locations of basal freeze-on occur throughout the GIS, including near Petermann Glacier where Bell et al. (2014) provide observational evidence of packages of basal freeze-on ice. Our model estimated locations (blue in Figure 7), and the Bell et al. (2014) observed locations (black in Figure 7) show some agreement and some disagreement. We interpret the disagreements as a combination of artifacts in the basal DEM and artifacts due to limitations in our routing model. We address each of these in the discussion section.

## 4.4   Basin-scale changes of VHD

Basin-scale changes between the three different time periods considered here are well illustrated when viewed as change in VHD per basin (Fig. 4) or percent increase in VHD per basin (Fig. 5). Basin size influences results for the former, and the effect is removed for the latter. Because integrated per basin VHD removes the effect of flow routing, VHD per basin is approximately proportional to runoff per basin, and changes in basin VHD are proportional to changes in basin runoff.

The difference between the Tedstone et al. (2015) reference period (TR) and their baseline period (TB, Fig. 4a) is a ~0.05 GW increase in the energy in each basin in the southwest sector where Tedstone et al. (2015) observed a general velocity

decrease. Elsewhere, increases were minimal (southeast) or negative. A similar pattern emerges between the historical and present cases (Fig. 4b), with the bulk of the change in the southwest sector, but larger than for TB-TR. At present there is a ~0.3 GW difference compared to the historical rate in the southwest sector. Between the present and the 2090s under the RCP 4.5 scenario, >0.3 GW increases occur in many sectors except the northwest (Fig. 4c). In the RCP 8.5 scenario, >2 GW increases occur in several basins (Fig. 4).

Percent increase between historical and present become larger with latitude. All of Greenland has experienced an increase, with many regions showing a 2- to 3-fold increase (+100-200%) in VHD (Fig. 5 panel P/H). Runoff, and therefore VHD, in the north of Greenland has experienced the largest percent increase. This is because VHD values are so small there that all increases appear large when viewed on a percentage scale.

However, the choice of baseline matters. The historical and present periods are 1960 – 1999 and 2010 – 2019 respectively. If the TR (1985 – 1994) and TB (2007 – 2014) periods are used instead, our results approximate the results from Tedstone et al. (2015), which showed a 50% increase in runoff in the basins just south of Jakobshavn Isbræ. The reason for the difference between a ~50% increase in runoff reported by Tedstone et al. (2015) and the ~10-20% increase in VHD obtained here may be because our TR data come from a future simulation and not a reanalysis, or because VHD is a function of runoff volume (the property reported by Tedstone et al. (2015)), but also the other terms in Equation 2.

## 4.5   VHD along a flowline in southwest Greenland

Viewing results along a flowline highlights that the hydropotential gradient driving the flow becomes larger and more variable toward the margin (Fig. 6a). Along a single subglacial hydrological flow line, step-increases in volume occur where other major tributaries join the flowline displayed here, causing 3 to 4 orders of magnitude increase in water volume (Fig. 6b). This increase in water volume leads to a corresponding increase in VHD (Fig. 6c). Variations in bed topography and ice thickness give rise to variations in the gradient of $\phi$ along the flowline (Fig. 6a) and, hence, to variations in VHD along the flowline (Fig. 6c). Although VHD generally increases toward the margins (Fig. 6c), due to increasing flux (Fig. 6b), the hydropotential gradient (Fig. 6a) appears as a high-frequency overprint on top of the background flux-driven signal, with 1-2 orders of magnitude change in VHD over just a few of the 150 m grid cells. If surface runoff is not routed to 2000 m elevation, then the flux begins a few hundred km inland with associated VHD inland. However, VHD under >2000 m of ice is less than GHF, even under RCP 8.5 scenario (Fig. 6c), and there is no significant change near the margin. Gaps in Fig. 6c are due to low gradients at those locations causing the release of only small amounts of VHD. Low gradients (Fig. 6a) occur naturally if the bed and ice surface are flat, or other combinations such that the LHS of Eq. (1) is small. Alternatively, low gradients may be due to actual high gradient flow paths below the model resolution. In this case, the neighboring upstream and downstream grid cell gradients are likely reduced, contributing to an increased gradient between the two, with the total gradient over three grid cells unchanged.

## 4.6   VHD and GHF

Frictional basal heating is up to 0.2 W m$^{-2}$ near the Russell and Leverett glaciers (Brinkerhoff et al., 2011), while geothermal heat flux (GHF) is estimated at ~0.050 W m$^{-2}$ (Shapiro and Ritzwoller, 2004) to as little as 0.030 W m$^{-2}$ (Meierbachtol et al.,

2015). The logarithmic scale used in Fig. 6c makes the differences between these heat sources and VHD appear small, but near the margin VHD exceeds GHF by one to two orders of magnitude. At present, VHD releases more heat than GHF from ~75 km up the flowline (< 75 km inland due to a sinuous path) to the margin. In the future, when larger volumes of water flow from farther in the interior, the zone where VHD surpasses GHF may increase its reach to ~150 km upstream from the ice margin (Fig. 6c).

## 5   Discussion

### 5.1   The impact of VHD

When discussing subglacial hydrology, a simplification can be made that water decreases basal friction and leads to faster ice sliding but VHD leads to conduit formation, reduced subglacial water pressures, and slower ice sliding. However, because VHD is generated by water flow, some condition is needed to define which behavior is dominant in a given setting.

Near the margin, steep hydropotential gradients (Fig. 6a) favor high VHD generation for a given water discharge. In addition, thinner ice reduces the creep closure rate. These two combine to form larger conduits that stay open longer. In the interior, smaller gradients lead to smaller conduits, and higher overburden pressure will collapse any openings.

Existing observations support the above hypothesis. The marginal zone ice response to VHD has been well-studied and observed in the southwest sector, where the summer increase in runoff is correlated with reduced glacier velocities (Bartholomew et al., 2010; Sundal et al., 2011; Tedstone et al., 2015). The interior ice response to increased runoff is less well-studied. However, Bartholomew et al. (2011) show that ice does not slow down later in the season as more runoff reaches the bed, and Doyle et al. (2014) shows a year-on-year increase in velocity even with increasing runoff.

### 5.2   Increasing runoff and VHD

As the climate warms in the future an increase in the supply of surface runoff to the bed will lead to an increase in subglacial VHD. We predict a 5-to-7 times increase in VHD by the end of the century under RCP 8.5. The impact of this increase is uncertain. This is because other results show that glaciers can either increase (Zwally et al., 2002) or decrease (Sundal et al., 2011; Tedstone et al., 2015) their mean annual velocity as additional water accesses the bed. The theory of efficient versus inefficient subglacial drainage explains the different observations, but it is not known what mode of subglacial water drainage currently exists beneath specific parts of Greenland, nor what specific thresholds may cause switches in drainage modes, nor the associated response of ice dynamics to these switches. We speculate that increasing VHD will have different impacts near the ice sheet margin as compared to the interior of the GIS, and that the well-studied southwest sector may not be representative of other regions since it is already the part of the GIS that is experiencing the highest VHD.

In the southwest sector, Mayaud et al. (2014) has bridged the gap spatially between the margin and the interior, and temporally between present and future, using the same runoff and RCP scenarios used in this study. They show that near the margin in the Paakitsoq region, conduits are likely to form earlier, remain longer, and reduce glacier velocity under RCP 4.5 and 8.5

compared to present. They also hypothesize that under thicker ice, conduits are unlikely to form, and increased water input into a more distributed subglacial drainage system may lead to an increase, rather than a decrease, in glacier velocity.

An increase in the interior ice velocity and a decrease in marginal velocity suggests that surface slopes and driving stresses will change, a result confirmed by Shannon et al. (2013). However, the range of possible results is not well enough constrained there to know the impact of the change in driving stress or how downstream ice may pull or block the flow from upstream (e.g. Ryser et al. (2014)). Our model uses a fixed surface under all scenarios, which means a constant hydropotential gradient. Margin retreat or changing hydropotential near the margin are not simulated.

Compared to the southwest sector the 2000 m contour (used here as an approximate boundary that defines where water accesses the bed) creates a much wider zone of active basal hydrology in the northeast and a much narrower zone in the southeast (Fig. 1). There are also much steeper bed elevation gradients on the east coast relative to the west and southwest coast (Fig. 2, $\nabla \phi_z$), leading to increased VHD potential per grid cell (Fig. 2, $\nabla \phi$). That increased potential is somewhat mitigated by less runoff on the east coast (Fig 1), but there is still predicted to be increased VHD in the east compared to the west per grid cell (Fig 3). However, east coast drainage basins are smaller, meaning the largest VHD increases per basin occur in the southwest sector (Fig. 4). Until field campaigns and model studies focus on regions of the GIS which have a different hydrological, basal, and basin system than the present southwest sector, the results of most southwest-focused studies should be generalized with caution.

### 5.2.1 Increasing runoff and VHD in the marginal zone

A threshold of a 50% increase in runoff has been identified by Tedstone et al. (2015) as leading to a widespread reduction in glacier velocity in the southwest sector marginal zone. Our results show a 10-20% increase in basin-cumulative VHD over the same region and time period used by Tedstone et al. (2015) (Fig. 5, right panel), which is equivalent to ~0.05 GW per basin (Fig 4a). Elsewhere, except for a region in the southeast and a few isolated basins in the northwest, most basins show a decrease between the TB and TR periods, but currently the regional velocity changes elsewhere are not known beyond the localized outlet glaciers. If the regional correlation between our results and Tedstone et al. (2015) shown here is causal, then an increase of VHD on the order of a 0.05 GW per basin may be near the threshold that causes a reduction in ice marginal zone velocities.

Under the RCP 4.5 scenario, nearly every basin will experience a 0.05 GW increase in VHD, with many gaining >0.5 GW. These significant increases in VHD should cause conduits, where they do form, to form more quickly and grow to larger dimensions than they do at present. At first, small increases in runoff may lead to faster ice flow, but eventually significant increases of VHD around all of GIS in the future may cause a slowdown in marginal zones. The above discussion is based on the current observed behavior of land-terminating glaciers, which have an observational bias to the southwest sector. At marine terminating glaciers, this effect may be less important than other processes determining glacier velocity and its variability, such as the processes related to ice-ocean interactions (e.g. Walter et al. (2012)). There may be fundamental differences in VHD between marine- and land- terminating glaciers. Relative to land-terminating glaciers, marine-terminating glaciers have a depressed PTT at the discharge location. Marine terminating glaciers may also have reduced surface slopes near their margin

and different basal topography (producing different hydropotential gradients), from the cumulative effect of a different flow regime due to their marine boundary.

### 5.2.2 Increasing runoff and VHD in the interior

Future runoff and VHD will be distributed over a longer part of a year relative to the present since climate warming prolongs the melt season in Greenland (Hanna et al., 2008). It will also be distributed spatially further inland relative to the present, and in the interior inefficient conduits are less likely to form (Dow et al., 2014). Additional heat and water at the bed will warm the basal ice. If the water cannot be evacuated efficiently by conduits, basal pressures will increase. Given that the primary cause of velocity decreases near the margin is assumed to be the evolution of subglacial conduits reducing basal water pressures, their absence in the interior means we expect velocities to increase, in line with existing observations (Doyle et al., 2014; Bartholomew et al., 2011). However, traditionally the relationship between basal friction and velocity is observed near the margins, where accelerating ice can flow unimpeded. If the interior ice accelerates due to increased basal pressures, but the marginal zone ice slows down due to increased VHD, then the interior ice flow may be modulated by the marginal ice flow (e.g. Ryser et al. (2014)).

When VHD occurs in new locations at the GIS bed it may convert a frozen bed to temperate and increase ice sliding (Parizek and Alley, 2004; Shannon et al., 2013). This is not likely to impact most of Greenland, where the bed is not frozen or the bed is frozen but predicted to remain isolated from new VHD. However, the northern sector has a frozen bed in regions where, according to our model, VHD increases markedly under the RCP 4.5 and 8.5 scenarios (MacGregor et al., 2016).

### 5.3 Other uses of energy than VHD

Not all of the incoming energy is converted to VHD and used to melt conduits, warm the basal ice, or warm the bed. The primary use of energy other than VHD is change in heat content of the water itself which needs to compensate for the spatially changing PTT. Classic glacier theory (i.e. Röthlisberger (1972)) states that when flow is driven by a pressure gradient, 35% of the available VHD is used internally to keep the water at the phase transition temperature (here termed a "loss"), and the remaining amount is dissipated as heat.

Our results show that different sectors may experience large changes in VHD relative to each other due to changes in the PTT. In practice, losses near 35% occur often - whenever elevation change across a grid cell is close to 0, and flow is driven primarily by a pressure gradient (Fig. 2). Negligible losses, near 0%, are also relatively common, when ice thickness does not change and both surface and bed elevation have similar gradients. Gains of 35% may also occur where the surface remains nearly flat and the bed drops drastically. Finally, in some locations an increasing PTT may consume 100% of the available VHD and basal freeze-on occurs. Although here we use the term "freeze" and display locations of freeze-on in blue (Fig. 3), these regions inject excess heat into the subglacial water and basal ice (not tracked in our model), due to the release of latent heat as water freezes (Alley et al., 1998; Bell et al., 2014).

There are some disagreements in the location of basal freeze-on between our model and Bell et al. (2014) observations. The largest area of disagreement occurs in the upper Petermann catchment (bottom right of Fig. 7). In this area, the model

does not estimate freeze-on within a few km of the observed basal ice packages. Conversely, in the northwest sector, several observational transects running east-west appear just southward of similar east-west model clusters of freeze-on locations. It seems likely that these agreements may also indicate an artifact in the basal DEM. The basal DEM is built, in part, from these same Bell et al. (2014) observational transects (Morlighem et al., 2014). The regular north-south spacing and linear east-west alignment suggests a processing artifact. Finally, our routing model treats over-deepenings and locations of basal freeze-on the same as other regions, which may be an invalid assumption. Hooke (1994) showed that on mountain glaciers, water preferentially routes englacially as it crosses an over-deepened section, rather than subglacially. Based on Hooke (1994), alternate basal paths may be used by the water to avoid locations favorable for freeze-on.

It seems likely that part of the cause for the disagreement between our model results and Bell et al. (2014) observations is because the basal DEM does not accurately represent the bed topography, at least at 150 m resolution. If this is the case, it impacts locations of basal freeze-on, and has some impact on the flow routes modeled here, but should not change the basin-scale results. Those results are primarily a function of the surface runoff volume and location, large-scale ice-thickness, and locations of the outlet glaciers. The path the water takes between the source and sink only impacts local VHD distributions, not basin-scale quantities.

### 5.3.1  Geothermal Heat Flux

GHF is expected to be more spatially uniform than VHD, at least near the margin where conduits concentrate the flow. GHF is temporally more steady than VHD, which primarily occurs when surface melt is active. Nonetheless, it is worth comparing the magnitude and distribution of the two. Historically the total VHD of 46 GW under the runoff area was similar to the total GHF of 35 GW in that same area. That is no longer the case in our calculations for the recent time period, and although GHF flux does not change, the integrated amount does change because the area of integration changes. By the end of this century under RCP 8.5, VHD will contribute 310 GW but GHF only increases to 44 GW due to a slight increase in the runoff area that reaches the bed, when surface runoff is routed to 2000 m elevation before moving to the bed. If the 2000 m limit is removed, then under the RCP 8.5 scenario the area of runoff nearly doubles in size (Fig. 2), and GHF increases from 44 to 81 GW, while VHD only increases from 310 to 324.

VHD and GHF comparisons and relative changes between present and future are most likely to matter in the region > 75 km upstream of the margin and where VHD is active. This is because the change here a) switches which term is dominant and b) is far enough inland that conduits are less likely to form (Dow et al., 2014), meaning VHD is more likely to be spatially uniform rather than concentrated in smaller regions.

VHD dominates other basal heating terms considered in some glaciological models (for example, Brinkerhoff et al. (2011)). Models show that the GIS is sensitive to its basal temperature, with small differences in GHF producing significantly different GIS growth scenarios (Rogozhina et al., 2012). Local GHF highs also coincide with onset of fast ice flow (Fahnestock et al., 2001). The results of our analysis and these GHF studies suggest that if VHD changes from 1 to 2 orders of magnitude less than GHF, to 1 to 2 orders magnitudes more than GHF, it will likely decrease the importance of GHF in modulating spatial dynamics of the ice sheet, at least within the zone of active basal hydrology dominated by surface water penetration to the bed.

## 5.4 Erosion and sediment transport

Large amounts of eroded material are also flushed out from under the GIS each year (Cowton et al., 2012). The erosion rates implied by the sediment flux are already several orders of magnitude above the background (>1000 year) erosion rates (Koppes and Marchant, 2009). Larger VHD leads to larger conduits, faster water flow velocity, and higher erosion rates and sediment transport capacity. Conversely, slow subglacial water flow does not have as much impact on erosion and sediment transport (Hodson et al., 2016; Gimbert et al., 2016). Five times the amount of water flowing along the GIS bed by the end of the century will likely increase sediment removal (Bogen and Bønsnes, 2003). If increased VHD and water at the bed of the GIS simultaneously cause sediment removal rates to increase while reducing glacier velocities (Tedstone et al., 2015) and therefore the production of sediment (Herman et al., 2015), the state of the bed may change over the coming centuries to millenia from potentially deformable subglacial sediments to rigid bedrock (Weertman, 1964; Kamb, 1970; Tulaczyk et al., 2000; Bougamont et al., 2014)

## 5.5 The impact of model spatial resolution on results

In addition to the various limitations to this model discussed throughout the text, here we address the limits of the spatial resolution. The model resolution is a 5x5 km grid for most of the analysis presented here, which means results are smoothed over that area. In reality, subglacial discharges occur approximately on the order of one every 5 km along the coast (Lewis and Smith, 2009). If a single conduit on the order of 10 – 100 m wide carries all of the water (Fig. 1) and is subject to all of the heating (Fig. 3), then values reported (here spread over 5000 m) are likely to be one or more orders of magnitude larger in focused regions, and one or more orders of magnitude smaller outside the conduit. This limitation of the model domain is less important in the interior, where large conduits are less likely to form.

Our treatment of en- and sub- glacial hydrology is simplified because it does not represent actual conduits, but is at the same time more comprehensive than in existing global climate or ice-sheet models (e.g. Pollard and DeConto (2012)). If dynamic basal pressure is not required by the model, this approach may offer a computationally efficient yet improved method to incorporate parameterizations of VHD at their existing grid resolution.

## 6 Conclusions

The high potential energy contained in large volumes of GIS surface meltwater is mostly dissipated as heat at the ice sheet bed. This dissipated energy averaged 46 GW each year between 1960 and 1999, but has recently increased to 66 GW and will likely increase to 110 or 310 GW by the end of the century under RCP 4.5 or 8.5 respectively. This viscous heat dissipation by subglacial water is the dominant basal heat source near the margin, and its impact will move inland due to increasing flux, even if conduits do not form in the interior. Under RCP 8.5, VHD will be about seven times larger than the 44 GW contributed by geothermal heat flux to the same area. That may decrease to four times larger if runoff penetrates to the bed at elevations >2000 m. In this case VHD does not change significantly, but the area used to integrate GHF nearly doubles in size (Table 1).

Up to 7 times additional future VHD at the ice sheet bed (relative to the historical amount) should result in a similar 7 fold increase in basal ice melt volume and is expected to contribute to more numerous, larger, longer-lasting, and more widespread subglacial conduits in the margin zone. Based on recent measurements by others and glaciological theory of ice sliding, increased VHD may decrease glacier velocity at the margin, and accelerate it in the interior where conduits either do not form or have insufficient impact on subglacial water pressures to influence ice sliding rates. The marginal decrease may be offset by other processes and there may still be a net acceleration, especially at marine terminating glaciers.

## Appendix A: About This Document

This manuscript is prepared with the intent to create a "fully reproducible" scientific publication. We may not have completely succeeded, but have made progress in this direction. In order to be fully reproducible at the binary-level, a clone of our operating system with the full analysis software should be provided. This could be done with a virtual machine (VM) but we have not taken this step because VMs require ~20 GB of space, and journals do not yet support this size of supplemental materials.

Instead, we used only free and open source software above the operating system level, document in detail the version(s) of all software packages used, and provide every line of code required to reproduce the document, beginning with the commands to download the MAR (Fettweis et al., 2013) and IceBridge BedMachine Greenland, Version 2 (Morlighem et al., 2015, 2014) data sets, followed by the GRASS GIS (Neteler et al., 2012) and Python commands to produce intermediary data products and graphics.

The supplementary data is a plain-text file that contains the manuscript text and all of the code. As plain text, it can be viewed in any editor or document viewer. However, its internal structure is that of an Emacs Org Mode (Schulte and Davison, 2011; Schulte et al., 2012) file and is best viewed in Emacs, which supports execution of the embedded code blocks. A reader should be able to reproduce the contents of this document, although it will require 3rd-party applications (GRASS, Python, etc.), and, optionally, a similar system-level Emacs configuration as the authors.

*Acknowledgements.* We thank D. van As for initial discussions on this topic, M. Morlighem and X. Fettweis for providing accessible and documented data, R. Bell for sharing data, and D. Pollard, anonymous referees, and The Cryosphere Discussion reviewers for comments. K. D. Mankoff was funded by NASA Headquarters under the NASA Earth and Space Science Fellowship Program (Grant NNX10AN83H) and the Postdoctoral Scholar Program at the Woods Hole Oceanographic Institution, with funding provided by the Ocean and Climate Change Institute. S. Tulaczyk was funded by NASA grant NNX11AH61G.

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

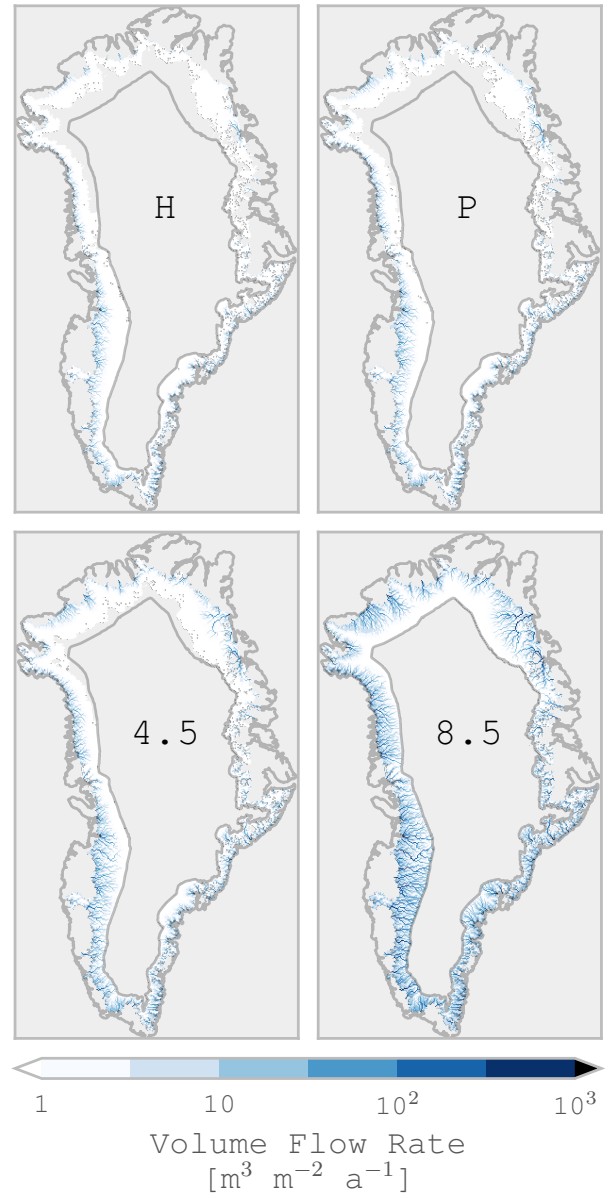

**Figure 1.** Accumulation of subglacial water flowing through each grid cell. Results are presented on a 5x5 km grid. Labels represent (H)istorical mean from 1960 – 1999, (P)resent mean from 2010 – 2019, (4.5) years 2090 – 2099 under IPCC AR5 RCP4.5, and (8.5) same as (4.5) but under scenario RCP8.5. Gray contours mark 0 and 2000 m elevation.

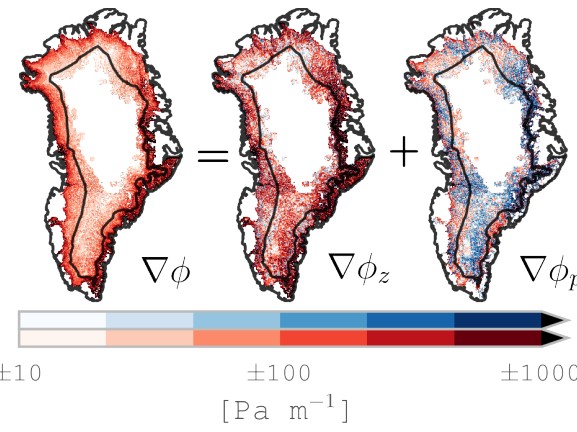

**Figure 2.** Net hydropotential gradient for each cell ($\nabla\phi$), and the decomposition of net hydropotential gradient to elevation-driven hydropotential gradient ($\nabla\phi_z$) and pressure-driven hydropotential gradient ($\nabla\phi_p$). Red implies flow downhill or under thinning ice. Blue implies flow uphill or under thickening ice. Black lines are sea level and 2000 m contour.

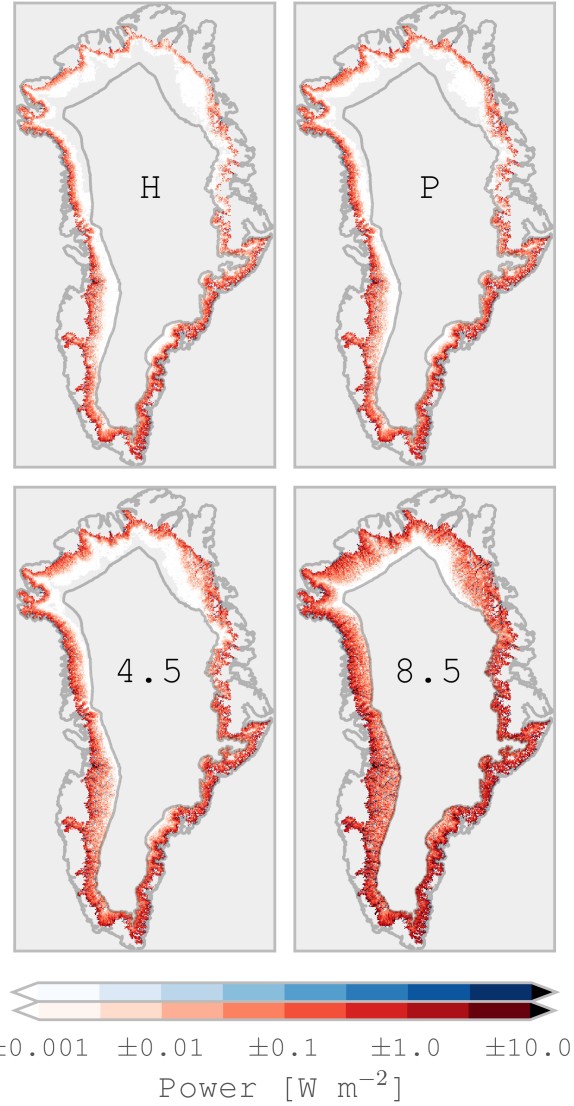

**Figure 3.** Heat released at the bed due to VHD. Labels same as Fig. 1. Colorbars represent heating (red, positive) and cooling (blue, negative). Gray contours mark 0 and 2000 m elevation.

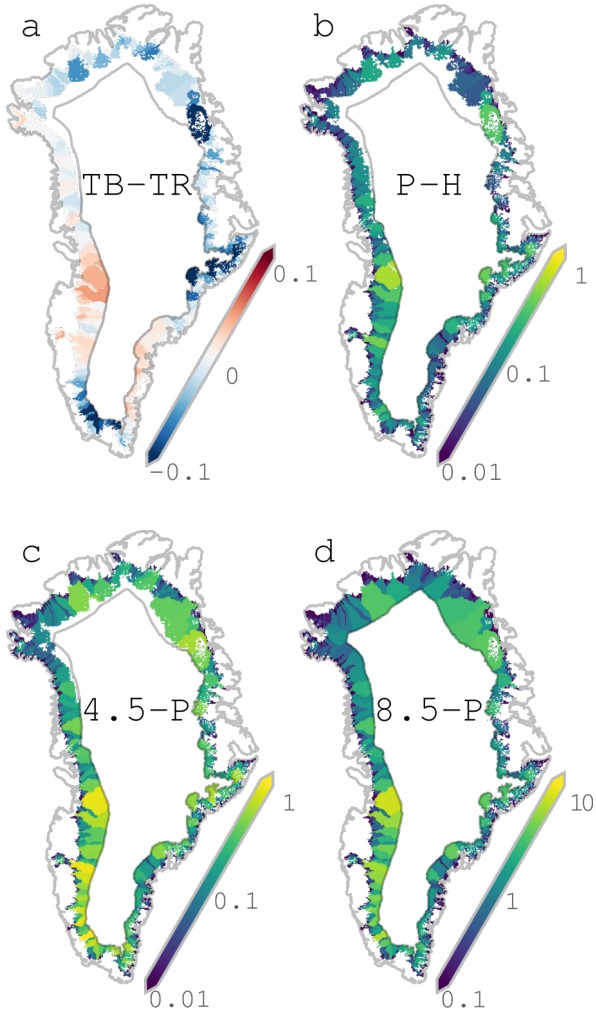

**Figure 4.** Change in GW ($10^9$ Watt) VHD per basin. Label TB-TR represents increase from reference to baseline periods (1985–1994 and 2007–2014) from Tedstone et al. (2015), P-H increase from historical to present, 4.5-P increase from present to 2090s under RCP 4.5, and similarly for 8.5-P. Gray contours are 0 and 2000 m elevation. Tedstone et al. (2015) study region highlighted with black contour line in panel (a). Scale is linear for TB-TR, logarithmic for all others, and ranges differ.

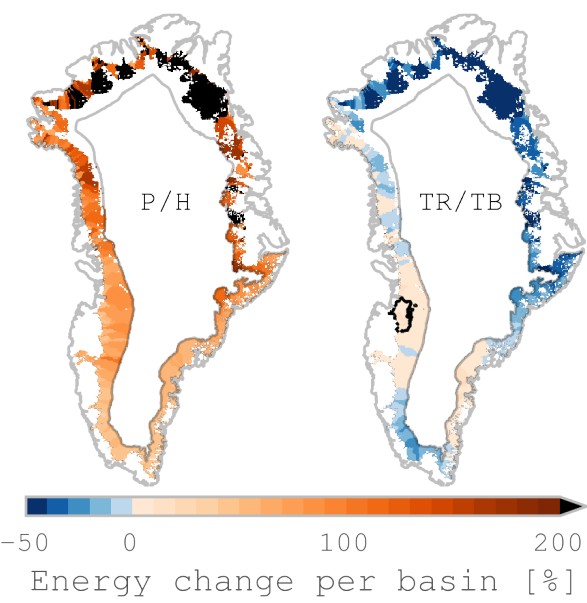

**Figure 5.** Relative change in VHD per basin highlighting the impact of different averaging periods. P/H is 100 x present divided by historical, and TR/TB is 100 x the reference (1985–1994) divided by baseline (2007–2014) years from Tedstone et al. (2015). Gray contours are 0 and 2000 m elevation. Tedstone et al. (2015) study region highlighted with black contour line right panel.

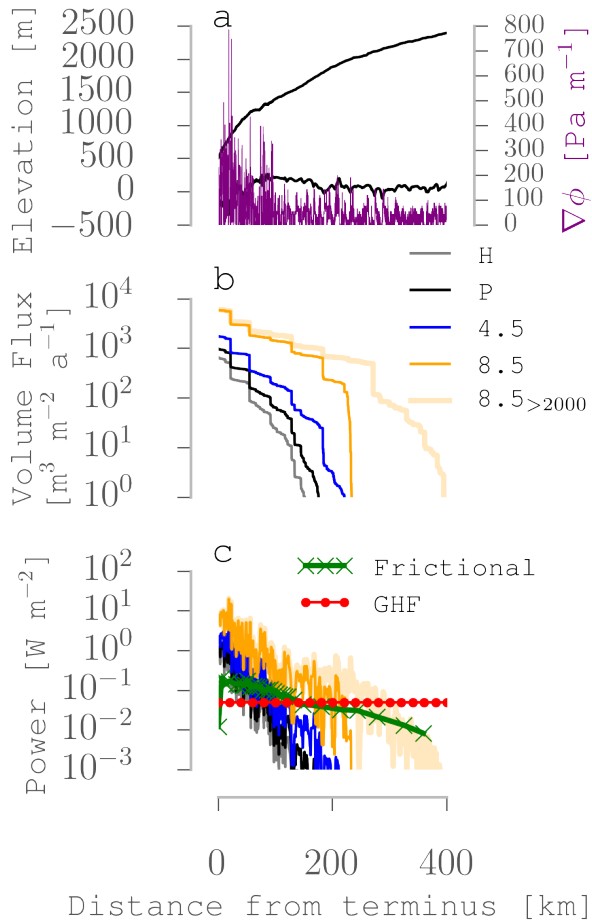

**Figure 6.** Detail along a flowline in Southwest Greenland. a) surface and bed elevation (left axis) and gradient of $\phi$ (right axis), b) flow rate of subglacial water, and c) power from VHD, frictional heating, and geothermal heat flux (GHF). Legend labels H, P, 4.5, and 8.5 same as Fig. 1. Label $8.5_{>2000}$ represents runoff that can access the bed at elevations >2000 m surface elevation. Frictional heating from Brinkerhoff et al. (2011), and GHF from Shapiro and Ritzwoller (2004). Lines in panel (c) are smoothed to reduce visual noise and are actually as variable as panel (a).

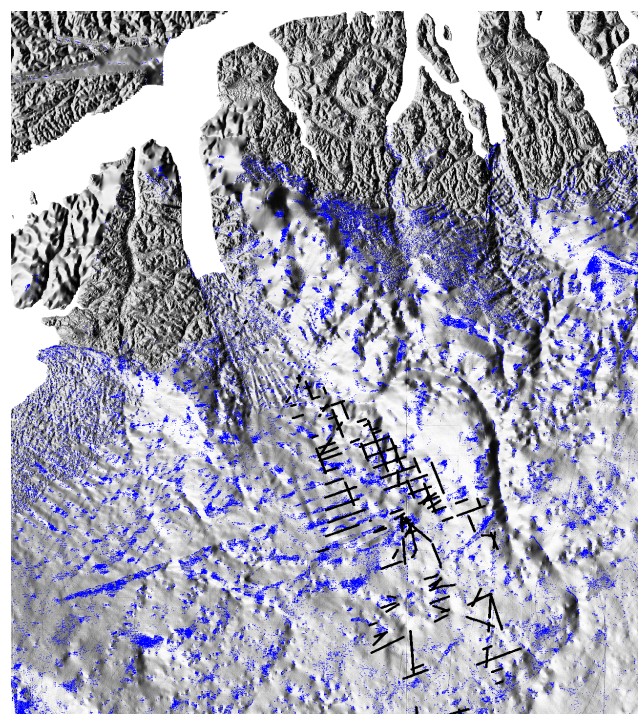

**Figure 7.** Close-up of Petermann Glacier region at 150 m resolution. Gray basemap is shaded relief of hydropotential gradient. Blue dots and black lines are locations of basal freeze-on predicted by the model and from Bell et al. (2014), respectively. Each blue dot is a 150 m x 150 m square pixel.

**Table 1.** Properties of Greenland runoff and viscous heat dissipation. (H)istorical period covers 1960 – 1999, (P)resent spans 2010 – 2019, and the RCP(4.5) and (8.5) periods span 2090 – 2099. Column $8.5_{>2000}$ shows model results when runoff can access the bed at all elevations. Runoff volume from MAR (Fettweis et al., 2013). Geothermal heat flux from Shapiro and Ritzwoller (2004) calculated only under runoff area.

| Property | Units | H | P | 4.5 | 8.5 | $8.5_{>2000}$ |
|---|---|---|---|---|---|---|
| Runoff volume | $km^3\ a^{-1}$ | 244 | 345 | 524 | 1278 | 1278 |
| Maximum discharge per 5x5 km grid | $km^3\ a^{-1}$ | 5 | 7 | 10 | 31 | 31 |
| Viscous heat dissipation | $10^9$ W | 46 | 66 | 110 | 310 | 324 |
| Geothermal heat flux | $10^9$ W | 35 | 36 | 40 | 44 | 81 |