# Peer review of "The past, present, and future viscous heat dissipation available for Greenland subglacial conduit formation"

_The Cryosphere, 2016_

## Referee Comment (RC1) · Anonymous Referee #1 · 24 Jun 2016

This paper calculates the energy content of supraglacial runoff from the Greenland ice sheet that is available for melting of subglacial conduits on its path to the ice margin. The paper discusses the energetics of water flow englacially and subglacially, and uses historical and future (model) estimates of runoff from the ice-sheet surface to estimate the consequent amount of viscous heat dissipation at the bed.

I am hesitant to recommend publication for two reasons. Firstly, I find the calculations rather simplistic, and am not sure there is anything new in it that is not already well understood. Secondly, I think there are some issues with the methodology, or at the least with its unnecessarily convoluted explanation in the manuscript. I address these in turn.

[Figure]

It is a central part of the established theory of subglacial conduits, due to Rothlisberger and Nye among others, that conduits enlarge through the transfer of gravitational potential energy, via viscous dissipation, to latent heat for melting the conduit walls. The rate at which this energy transfer occurs is proportional to the volume flow rate of water and the rate of change of hydraulic potential with distance, with corrections for the advection of heat required to keep the water at the pressure melting point. Changes in kinetic energy are relatively small compared to these other terms, and most estimates except in volcanic environments suggest that the excess heat in the water above the pressure melting point is also negligible. These facts are included in most models of subglacial hydrology, and lead to the often quoted result that a larger volume flux of water causes increased melting of the conduit walls and can therefore be driven out from under the glacier by a smaller potential gradient, despite the larger flux. The fact that increased runoff is therefore expected to lead to larger conduits is quite well established, and the fact that this is due to increased viscous heat dissipation goes without saying. It seems to me that the results of this study essentially re-express the modelled increase in surface runoff (from other studies) in terms of gravitational potential energy. There is a modification for the pressure melting effect, but this is nonetheless the essence of what is done. Perhaps some readers may find it helpful to have the runoff increase re-expressed in this way, but I did not find it particularly illuminating.

Secondly, I found some aspects of the presentation confusing. It was not totally clear what calculation was actually performed to produce the final numbers. As far as I can work out, the energy available for viscous heat dissipation at the bed is the change in gravitational potential at the basal elevation from where the runoff reaches the bed to where it leaves, plus 90% of the gravitational potential energy due to the ice thickness where the runoff reaches the bed (the first 10% being assumed lost to the atmosphere), minus the fraction $C\_T*rho*c\_p$ ($\sim$0.3) of this potential due to ice thickness (which is required to keep the water at the melting point). Couldn't it be explained in a paragraph or with one explicit equation?

The part of this that I am most uncertain about is the gravitational potential taken at the outflow - in section 3.6 it suggests that this is taken to be sea level, which seems quite a simplification. If this is what is done, what was the point of the hydraulic potential routing? I would have thought that such routing tells you what elevation the water leaves the ice, and that making an ad-hoc assumption about what elevation it leaves at is therefore unnecessary (though using sea-level is correct for any outflow below sea-level, where the hydrostatic 'pressure energy; cancels the negative gravitational potential energy).

There seemed to be a lot of repetition in the methods description, particularly about the pressure melting point. The comment in section 3.2 that the influence of the pressure melting effect is not 'zero sum' may be true if one is concerned with the energy balance only at the bed, but it must be zero sum when considering the energy balance over the whole path of the water from ice surface to margin, which is what seems to be under consideration in other parts of the manuscript (eg figure 1). If the water descending through the moulin has cooled and lost sensible heat, that heat must have gone somewhere (presumably into melting the surrounding ice, as in the subglacial conduits).

Figure 1 is rather confusing. What is meant by 'amount of energy' as shown in the bars? Is this the energy of a fixed mass of water, and if so how do you account for the energy used to increase the mass of water (from melting the conduit walls)? I could not understand the meaning of the phase transition temperature in this diagram - how is this expressed as an energy (yellow hash)? It seems to increase from the top to the bottom of the moulin (during pressurisation) and then to increase again on the passage to the margin (during depressurisation). This cannot be right as the change is in the opposite sense (the temperature decreases during pressurisation, and increases during depressurisation). Similarly, there is some confusion about gravitational potential energy, which should continue to decrease (becoming negative) in the moulin as the water goes below sea level, exactly cancelling out the 'overpressure'.

[Figure]

Specific comments

Equation (1). Why use gamma here, and rho elsewhere, and in some places m. Couldn't the notation be made more consistent?

Equation (3). What is $z_o$ taken as here? Potential energy always needs to have a reference point, but it does not make sense to have a different reference point for different parcels of water (as seems to be the case here, since elevation at the terminus would vary) - comparison of potential energies is then meaningless.

Equation (6). Delta phi here is a change in potential, but when plotted in the figures it seems to be a gradient (with units Pa/m). Which is it?

Figure 6. Why are there gaps in the data in panel c? There are also clear issues here with taking a numerical gradient of the discrete data, and some form of smoothing to calculate the gradient might yield cleaner results.

---

## Author Comment (AC1) · 29 Jun 2016

We thank Anonymous Referee #1 (AR1) for providing insightful reviews. We respond here to the two reasons AR1 is hesitant to recommend publication. We summarize these reasons as (1) nothing new and (2) convoluted explanation of methodology.

For (1), we agree with AR1 that the calculations follow a long-established methodology for treating energy balance during water flow through glacial systems, but we ask for clarification about the statement from AR1 that there is nothing new presented here. Our intent was not to provide new insights into the theory of subglacial hydrology, but rather to apply the well-established theory to the entire Greenland ice sheet over three time frames spanning from recent conditions to the predicted future state of the ice

sheet.

There are many theoretical papers modeling subglacial hydrology (from Röthlisberger (1972), Weertman (1972), or Nye (1976) to Schoof (2010), Hewitt (2011) or Mauro Werder et al. (2013)). There are fewer papers examining energetics of meltwater drainage in Greenland, and the few existing ones are focused specifically on the present and the southwest sector of Greenland. Only a few that we know of address how it may change in the future (for example, Banwell et al. (2013) and Mayaud et al. (2014), cited in our Introduction section). We are not aware of any that consider future subglacial hydrology outside of the southwest sector of the Greenland ice sheet.

If AR1 has a reference to a paper, or papers, that discusses quantity, effects, and potential impacts of changing subglacial hydrology historically, at present, and in the future, for all of Greenland, we would greatly appreciate it if AR1 can provide such reference(s) as it appears that we may have missed some highly relevant publications.

For issue (2), we can simplify the explanation of the methodology. Although the energy balance of water flow has been resolved a long time ago and we are not trying to change this we, did not think that simply citing Röthlisberger (1972) was sufficient, and opted to include a more detailed methods and assumptions section.

Responding to some other points raised by AR1, we note that:

+ The results of this study do not "essentially re-express the modelled increase in surface runoff [...] in terms of gravitational potential energy". Our results express the modelled increase in surface runoff in terms of J and W/mˆ2 at the bed, after accounting for along-path energy losses and flow routing.

+ We can clarify our treatment of the outflow elevation and remnant gravitational potential energy due to that elevation when outlets are above sea level. In section 3.6 we state, "...ideal scenario of a flat bed and outflow at sea level". In the full analysis we do know and use the outflow elevation. The point of the hydrology routing is that

the spatial changes in elevation and thickness matter inland of the outflow location, as these changes induce spatial variability in heat released at the bed.

+ AR1 is correct that the pressure melting point is not zero sum, but we explicitly state that this manuscript is only focused on heating at the bed, as they point out in the comment.

We are grateful for the comments from AR1 and agree with the remaining issues raised by AR1. We will address them fully if given an opportunity to revise this manuscript.

References

Banwell, A. F., I. C. Willis, and N. S. Arnold (2013). "Modelling subglacial water routing at Paakitsoq, W Greenland". Journal of Geophysical Research: Earth Surface. (118), 1282–1295. doi: 10.1002/jgrf.20093.

Hewitt, I. J. (2011). "Modelling distributed and channelized subglacial drainage: the spacing of channels". Journal of Glaciology. 57 (202).

Mayaud, J. R., A. F. Banwell, N. S. Arnold, and I. C. Willis (2014). "Modeling the response of subglacial drainage at Paakitsoq, West Greenland, to 21st century cli- mate change". Journal of Geophysical Research: Earth Surface. 119 (12), 2619–2634. doi: 10.1002/2014JF003271.

Nye, J. F. (1976). "Water flow in glaciers: joÌĹkulhlaups, tunnels and veins". Journal of Glaciology. 17 (76), 181–207.

RoÌĹthlisberger, H. (1972). "Water pressure in intra- and subglacial channels". Journal of Glaciology. 11 (62), 177–203.

Schoof, C. G. (2010). "Ice-sheet acceleration driven by melt supply variability". Nature. 468 (7325), 803–806. doi: 10.1038/nature09618.

Weertman, J. (1972). "General theory of water flow at the base of a glacier or ice sheet". Reviews of Geophysics. 10 (1), 287–333.

Werder, M. A., I. J. Hewitt, C. G. Schoof, and G. E. Flowers (2013). "Modeling channelized and distributed subglacial drainage in two dimensions". Journal of Geophysical Research. 118, 2140–2158. doi: 10.1002/jgrf.20146.
* * *

---

## Referee Comment (RC2) · Anonymous Referee #2 · 15 Jul 2016

General comments

This work examines how specific climate change scenarios will increase the delivery of runoff-sourced gravitational potential energy to the bed of the Greenland Ice Sheet, which in turn will intensify formation of subglacial conduits there. The extent and size of conduits have (potentially profound) implications for ice velocity, which this manuscript touches upon but does not extensively discuss. Rather, the manuscript focuses on the assumptions and setup of the calculation. Its main conclusions are that gravitationally sourced heat dissipation exceeds geothermal flux and that conduit formation is expected to increase in future scenarios. These conclusions suggest the potential for slower ice velocity and enhanced sediment removal in land-terminating regions under

future climate scenarios.

I found the idea behind this manuscript to be worthy and novel. However, the manuscript suffers from a lack of focus and is bottom-heavy, with a very involved description of assumptions (many of which are standard in subglacial hydrology) and the setup of the calculation, that builds to hesitant and limited larger arguments. These factors together limit the manuscript's publication readiness. I do think that the central idea is worthy and could be better highlighted, and its implications more thoroughly discussed, in future drafts.

The manuscript is quite light on the implications of its calculation. In particular, it would benefit from a more thorough discussion of the effects of increased runoff on ice velocity; as it is now, this treatment is brief (Introduction only; it should be moved entirely to the Discussion) and greatly oversimplified. The authors state that increased runoff "appears" (P2 L6) to lead to overall slowdown, but this is based on only two cited studies (Sundal et al., 2011 and Tedstone et al., 2015), both of which focus on beds under lower-elevation ice (s<~1200 meters). The alternative (i.e., speedup) is not mentioned, even though this is an ongoing topic of discussion in the community. For instance, multiple studies (e.g., Bartholomew et al., 2011 and Shannon et al., 2013) contrast with the cited studies, especially farther inland (s>~1700 meters) from the ice margin (e.g., Doyle et al., 2014), where subglacial flow currently does not channelize and may not reach the "runoff threshold" required for channelization (Mayaud et al., 2014; cited in this manuscript). Because this manuscript considers subglacial conduit formation at these areas of the bed (s<2000 meters) as well as those near the margin (where the cited work is more applicable), it is crucial to recognize these distinctions and the current uncertainty in the runoff / speedup literature.

With a more mature discussion related to ice velocity (or additional discussion of other implications hinted at; e.g., sediment evacuation), a reduction and better organization of the methods and assumptions sections, and attention to the points below (and those from AR1, which I largely agree with), I think this manuscript could make a good contribution.

General points

A good deal of the text in Section 3 (the details of each assumption) is unwarranted. These are standard assumptions that require a few citations, not multiple detailed paragraphs.

PTT and energy conservation. It took me some time to work out why the authors claimed that this _not_ zero-sum. My best understanding is that the PTT accounting begins at the bottom of the moulin (P5 L6), where the PTT is below zero. To cool to that temperature, the ice released sensible heat into the moulin (i.e., into the englacial, not subglacial, system). That sensible heat is, I think, part of the ∼35% of the gravitational potential energy that is lost (i.e., not transferred to basal ice). This needs to be made much clearer; perhaps with adaptations to Figure 1 (discussed later in this review).

By including the effects of the pressure-dependent melting point and using a relatively fine (5x5-km) grid, the authors obtain a detailed result of areas of expected basal freeze-on and of enhanced conduit formation. This has potential for application to local studies (and this appears to have been tentatively investigated, e.g. comparison to Bell et al. (2014), although without great success), or for regional assessments (Northeast, Southwest, . . .) as commonly done in Greenland. Yet the authors discuss only ice-sheet-wide totals; I am not convinced that more than the back of an envelope was really needed for this.

I was distracted by the authors' use of numeric examples with multiple intermediate steps to demonstrate simple arguments. It would be simpler, more compact, and more elegant if the authors were to state the expressions they wish to compare, and then plug in the physical constants. For instance, P6 L18-20 is easily expressed as PE / KE = 2 g h / v^2 = 200 for the stated h and v. Another example is with PTT on P5 L2-5.

Specific points

P1 L13-14 This sentence is difficult to parse

P2 L6-9 This sentence is also unnecessarily complex, with multiple negatives, and contains two thoughts (descriptions of two studies) spliced together. Fundamentally, the logical flow is backwards, with the conclusion presented first and the evidence trailing behind.

P2 L13 data _are_

P2 L18 Mayaud et al. (2014) studied Paakitsoq, not Russell Glacier.

Regarding Mayaud et al. (2014): That study is a nice introduction to this manuscript and should also appear in the Discussion, not just the Introduction. That paper makes an explicit link between increased conduit formation and ice velocity that is only summarily discussed in this study; the authors could (and should) leverage this to make a more convincing argument about future ice velocity.

P2 L25-26 Pollard and DeConto (2012) – Similarly, this citation and future application belongs in the Discussion or Conclusion, not the Introduction.

P2 L28 extra word "a"

P3 L3 capitalize Glaciers

P3 L4 This section could use a short introduction that explains that all terms of the Bernoulli equation and all stages along the water's path will be examined to make the best approximation. The authors do part of this in P3 L14-17, which should be moved earlier in the section and emphasized.

P3 L11-13 I found these two sentences to be a bit pedantic. At the very least, they are unnecessary because these terms are never again used in the manuscript.

P3 L22 Figure 2 shows energy flux, not energy.

P3 L24 Need to specify "in situ" for this statement to be correct.

P3-4 L28-3 The elevation $z_s$ is set to a maximum of 2000 meters, even if the melt forms above there. However, in RCP8.5, runoff forms as high as 2800 meters (Figure 2d; South Greenland summit), which would contribute an additional 800 meters (+30%) of head. I would expect that when weighted with the small runoff volumes at high elevations, this addition would be small – but the effects of this approximation could be mentioned.

P4 L8-9 This seems ancillary. I would encourage the authors to drop it or weave it in better.

P4 L13 This reference is for alpine glaciers, which is fine, but this, and its applicability to Greenland, should be pointed out.

P4 L18-19 The second clause is redundant with the preceding sentence.

P4 L20 dissipates

P4 L21 I would like to see some justification for why this heat is "lost", presumably to the atmosphere (as suggested in the legend of Figure 1). Does significant wind pumping really occur at 50-100 meters depth inside a moulin? Alternatively, if this heat is "lost" to melting the moulin walls, this should be stated.

P5 L2 This is not an equation but an expression.

P5 L7 This sentence suggests that this is not done for marine glaciers, although I believe it does need to be done (in an integrated-displacement sense, the water parcel _does_ decompress to atmospheric pressure).

P5 L18 The link to the Supplemental Material is broken.

P5 L24-30 The logical ordering here is flawed. The explanation for the 1/3, 2/3 values needs to be explained up front before the results are presented. As it is, the 1/3, 2/3 numbers appear mysteriously and add confusion until they are explained a few sentences later.
P6 L 7-8 Choose either "downstream" or "down-stream" (I believe the former is more standard)

P6 L13 Strictly speaking and if you are willing to be arbitrarily precise, this is true of _all_ water.

P6 L25-27 Neither is this sentence true (due to hydrostatic pressure energy; see comment above for P5 L7) nor does it belong in this specific subsection.

P7 L1-14 is way overblown. Furthermore, it reads like a response to a review, not like a journal article.

P7 L18 Really do not need these details.

P7 L19-20 It is unclear why this is relevant.

P7 L27 "by the second law of thermodynamics" is completely unnecessary

P8 L2 A "conjecture" is not a very convincing statement.

P8 L5 "heating" here could refer to the water heating the ice, whereas I believe the authors are referring to sensible heating (PTT) of the water itself.

P8 L12-13 The figure does not show different units; both color bars show total annual energy flux. The only difference is /yr versus /sec. The "energy" unit stated in this sentence is incorrect.

P8 L30-31 It is unclear what error the instantaneous heat transfer assumption is introducing. And "because due to" is a typo.

P9 L4-5 capitalize Russell and Leverett Glaciers; is estimated

P9 L8-9 The location of GHF = VHD appears to be coincident with the s=2000 meters location, the elevation where the authors first allow runoff to penetrate to the bed. This may be a coincidence; however, it does seem that the penetration contour would be a strong control on the amount of VHD at inland locations. How sensitive is this result to

using s=1500 meters penetration contour, or a gradual ramp-up (i.e., 100% of available melt penetrating at s=1500 meters, 50% at s=1750 meters with the other 50% being routed downhill, and 0% penetrating / 100% routed downhill at s >= 2000 meters)? If the relative contribution of GHF versus VHD continues to be a primary conclusion of the study, this is a worthwhile exercise to include.

P9 L20-22 As far as I can reason, I believe this conclusion is backwards. If the authors' model included conduits, this would increase VHD near conduits, but at the expense of VHD at locations without conduits. Since conduits are more likely to form near the terminus than the interior, this inclusion would draw VHD _away_ from the interior.

P9 L24 What specific components of flow routing and VHD are the authors referring to? Obviously, flow routing (a path) and VHD (a quantity) are not the same.

P9 L28-29 Probably the authors mean that the DEM is not _accurately_ representing the topography.

P9 L31 How far, on average, are the Bell et al. (2014) basal freeze-on packages from where these results predict?

Regarding areas of basal freeze-on: This water will no longer follow subglacial hydropotential gradients, but instead surface gradients because it is attached to the ice. These are not always the same directions. This may or may not be important (likely it is not), depending on what fraction of water gets frozen on. This would be worth addressing briefly.

P10 L4 "conduits systems" typo; "non-conduit region" typo

P10 L7-10 These sentences are awkward. Is it always the same grid cell? What are "such locations", probably marine-terminating glaciers with large catchments?

P10 L11 These assumptions have already been stated in the Methods and Assumptions section, which is a more appropriate place for them than the Discussion.

P10 L13 This should be phrased something like "the basally sourced meltwater carries away the initial gravitational potential energy of the runoff, in the form of latent heat, as it exits the subglacial system." The water itself is not heat.

P10 L15-16 This sentence is interesting and important, and should be more prominent and/or pointed.

P10 L17-18 I do not think that any evidence for this statement has been presented. Also, "percentages" is vague – percentages of what?

P10 L21-22 According to Dow et al. (2014) (cited elsewhere by the authors), the subglacial conduit network is _unlikely_ to expand significantly inland.

P11 L2 Should specify "over land-terminating ice".

P11 L7 Should specify "total GHF integrated over the runoff area". As it reads now, it sounds like the solid earth will be warming in RCP 8.5.

P11 L10-13 This is a smorgasbord of facts. Needs better organization, logical flow, and build-up to the main idea.

P11 L19 are

P11 L21 cause

P11 L24-25 This idea is interesting, but the evidence the authors present suggests millennial timescales, not < 1 century.

P11 L27 missing "a"

P12 L2 missing "orders of [magnitude]"

P12 L6 Not just the ablation zone: the area with s<2000 m includes parts of the wet snow zone

P12 L8 Not "will become", as the authors have shown that it is already the dominant basal heat source (both P and H scenarios)

P12 L8 "swamp" seems a bit informal

P12 L28 its

Figures

Figure 1: The info is very small relative to the rest of the space, and the short definitions given to each colored box are inscrutable without reading the manuscript carefully. It is unclear where the water travels and why the bar can sometimes exceed 1. I had a very difficult time with it, and there are still components that I do not understand (what do the authors mean by "latent heat"? presumably that refers to VHD? why not call it VHD?). As a first figure, it is so inscrutable that it will drive away all but the most invested readers.

Figure 1 presents results, whereas it is customary for the first figure to illustrate the setup. I think this figure would benefit from a major overhaul, with the results removed and a focused inclusion of methods components, such as the path of the water, the transfer of VHD into / out of the ice sheet along an undulating bed, input / output / conversion points for the various terms (gravitational PE, PTT), etc. I do not think anything would be lost from the manuscript if the colored bars were removed entirely.

Figure 2: A negative sign is missing from the color bar label (W m^-2)

Figure 4: The units here are a bit confused. Delta-phi should be in Pa, as is phi. If the authors have divided by the 5-km distance between grid cells, that is Grad-phi and has the units shown on the color bar (Pa / m).

Figure 6: I agree with AR1 that the numerical gradient issues must be addressed.

Panel a: see units (Pa versus Pa / m) comments on Figure 4

Panel b: Some of the lines show nonzero subglacial flow inland of the s=2000 meter contour (x ∼ 140 km), which the authors defined as the upper limit for melt to reach the bed. What is the reason for this?

Caption: should specify that b) is the flow rate _of subglacial water_

Figure 7 caption: use parallel definitions for the three cases; i.e., either all three are "the increase from A to B" or all three are "the difference between B and A". Also "Joules".

Panel c: The symbol that the legend refers to is unclear.

References

Bartholomew, I. D., P. Nienow, A. Sole, D. Mair, T. Cowton, M. A. King, and S. Palmer (2011), Seasonal variations in Greenland Ice Sheet motion: Inland extent and behaviour at higher elevations, Earth and Planetary Science Letters, 307(3-4), 271–278, doi:10.1016/j.epsl.2011.04.014.

Doyle, S. H., A. Hubbard, and A. Fitzpatrick (2014), Persistent flow acceleration within the interior of the Greenland ice sheet, Geophys. Res. Lett., 41, 899–905, doi:10.1002/2013GL058933.

Shannon, S. R. et al. (2013), Enhanced basal lubrication and the contribution of the Greenland ice sheet to future sea-level rise, Proceedings of the National Academy of Sciences, 110(35), 14156–14161, doi:10.1073/pnas.1212647110.

---

## Referee Comment (RC3) · Anonymous Referee #3 · 27 Jul 2016

In this manuscript, the authors calculated the viscous heat dissipation (VHD) generated as a result of runoff reaching the bed of the Greenland ice sheet, for the past and present, as well as for two future climate scenarios. The main findings are that VHD is becoming an increasingly large component of the basal heat budget – which is expected to contribute more significantly to subglacial conduits opening in the future.

I find the results novel and interesting, and a valuable addition to existing related work. However, the clarity of the text must be improved throughout, as the main or important points are often lost with too many details / repetitions / confusing sentences. Overall, I agree with comments aimed at clarifying the text, as given by AR1 and AR2. Below, I give a few more specific points below.

[Figure]

General points:

The discussion on the influence of subglacial hydrology and conduits formation on ice velocity (in particular under future scenarios), is over-simplified in the introduction and discussions (also pointed out by AR2). The overall effect of increased meltwater delivery to the bed of the ice sheet is unresolved. Some work suggest net deceleration (as already discussed), but other suggest a possible net acceleration (e.g., Bartholomew, NatGeo 2010; Doyle,GRL 2014). As the main implication from increased VHD concerns subglacial conduits formation, the authors should develop the potential implication of their findings more thoroughly.

Specific points:

P3 L12: define m

P3 L24: likElihood

P3 L27: suggest removing "(1-2 grid cells in our models)" – this is specified again later.

P4 L29: Bring "these processes" to the same place in the text (water captured by crevasses and...?).

P7: overall way too long, and hard to follow. What are the key points?

P8 L13-14: Suggest replacing the sentence with a recall of Eq. 2.

P8 L15: would write "...and 14.3 EJ year-1 (with 1EJ=1x1018J)"

P8 L17-18: last sentence not necessary in my view.

P8 L 30: remove "because"

P9 L25: sentence could be simplified – I find the use of statement such as " V times Eq (6)" clumsy.

P9 L31: "perfect line up" between model and observations are rare, but it sounds like you were expecting it. It would be more useful to state how far apart the freeze-on

packages are, and state where uncertainties might be coming from. Do you expect the errors from the model to relate to its physics, or input (GHF distribution, runoff distribution etc…)? … also, the advection argument seems far-fetched.

P10 L11-13: very long sentence, the point is lost.

P10 L14: numbered repeated from paragraph above. Suggest that section is re-organized to avoid repetition.

P10 L32: Use EJ

P11 L1-2: This statement should at least be moderated, or could be removed, as this is an argument made (in a much more balanced way) in the conclusion.

P11 L7: sentence describing the increase in GHF is not clear.

P12 L10: missing "and" after parenthesis

Figures:

Figure 1: Agree with AR1 and AR2, the bars and infos are vey small. Re-drawing with larger bars would help, as well as explicitly showing where VHD comes into the picture.

Figures 2 -3 -5: Not sure if there would be space for this, but I feel like these would beneficiate from being enlarged, e.g., as a 2 lines / 2 columns panels presentation. This is particularly true for Figure 5, where it is very hard to see any freeze-on.

---

## Author Response (AR1)

**Reply to Reviewer 1**

Ken Mankoff and Slawek Tulaczyk

Replies from the authors are inline in normal font and differentiated from the reviewer comments in bold colored font.

**This paper calculates the energy content of supraglacial runoff from the Greenland ice sheet that is available for melting of subglacial conduits on its path to the ice margin. The paper discusses the energetics of water flow englacially and subglacially, and uses historical and future (model) estimates of runoff from the ice-sheet surface to estimate the consequent amount of viscous heat dissipation at the bed.**

**I am hesitant to recommend publication for two reasons. Firstly, I find the calculations rather simplistic, and am not sure there is anything new in it that is not already well understood. Secondly, I think there are some issues with the methodology, or at the least with its unnecessarily convoluted explanation in the manuscript. I address these in turn.**

We have rewritten the calculations and clarified the methodology.

**It is a central part of the established theory of subglacial conduits, due to Rothlisberger and Nye among others, that conduits enlarge through the transfer of gravitational potential energy, via viscous dissipation, to latent heat for melting the conduit walls. The rate at which this energy transfer occurs is proportional to the volume flow rate of water and the rate of change of hydraulic potential with distance, with corrections for the advection of heat required to keep the water at the pressure melting point. Changes in kinetic energy are relatively small compared to these other terms, and most estimates except in volcanic environments suggest that the excess heat in the water above the pressure melting point is also negligible. These facts are included in most models of subglacial hydrology, and lead to the often quoted result that a larger volume flux of water causes increased melting of the conduit walls and can therefore be driven out from under the glacier by a smaller potential gradient, despite the larger flux. The fact that increased runoff is therefore expected to lead to larger conduits is quite well established, and the fact that this is due to increased viscous heat dissipation goes without saying. It seems to me that the results of this study essentially re-express the modelled increase in surface runoff (from other studies) in terms of gravitational potential energy. There is a modification for the pressure melting effect, but this is nonetheless the essence of what is done. Perhaps some readers may find it helpful to have the runoff increase re-expressed in this way, but I did not find it particularly illuminating.**

We agree the theory presented here is well established. We do not know of existing publications which apply this theory to all basins in Greenland under RCP scenarios.

If the reviewer has a reference to a paper, or papers, that discusses quantity, effects, and potential impacts of changing subglacial hydrology historically, at present, and in the future, for all of

Greenland, we would greatly appreciate it if AR1 can provide such reference(s) as it appears that we may have missed some highly relevant publications.

**Secondly, I found some aspects of the presentation confusing. It was not totally clear what calculation was actually performed to produce the final numbers. As far as I can work out, the energy available for viscous heat dissipation at the bed is the change in gravitational potential at the basal elevation from where the runoff reaches the bed to where it leaves, plus 90% of the gravitational potential energy due to the ice thickness where the runoff reaches the bed (the first 10% being assumed lost to the atmosphere), minus the fraction C_T\*rho\*c_p (0.3) of this potential due to ice thickness (which is required to keep the water at the melting point). Couldn't it be explained in a paragraph or with one explicit equation?**

We have clarified the text. It remains more than one paragraph, and now 2 equations, but is still significantly clarified.

**The part of this that I am most uncertain about is the gravitational potential taken at the outflow - in section 3.6 it suggests that this is taken to be sea level, which seems quite a simplification. If this is what is done, what was the point of the hydraulic potential routing? I would have thought that such routing tells you what elevation the water leaves the ice, and that making an ad-hoc assumption about what elevation it leaves at is therefore unnecessary (though using sea-level is correct for any outflow below sea-level, where the hydrostatic 'pressure energy; cancels the negative gravitational potential energy).**

The elevation of the discharge is based on the bed elevation, and is therefore always above zero if land terminating or below zero if marine terminating. The example text that caused the uncertainty for the reviewer was preface by the phrasing, "...in the ideal scenario...".

**There seemed to be a lot of repetition in the methods description, particularly about the pressure melting point. The comment in section 3.2 that the influence of the pressure melting effect is not 'zero sum' may be true if one is concerned with the energy balance only at the bed, but it must be zero sum when considering the energy balance over the whole path of the water from ice surface to margin, which is what seems to be under consideration in other parts of the manuscript (eg figure 1). If the water descending through the moulin has cooled and lost sensible heat, that heat must have gone somewhere (presumably into melting the surrounding ice, as in the subglacial conduits).**

Repetition removed. We still do not zero-sum the PTT because our model is only basal. This distinction, and the model beginning at the bed with a depressed PTT, is clarified in the revision.

**Figure 1 is rather confusing. What is meant by 'amount of energy' as shown in the bars? Is this the energy of a fixed mass of water, and if so how do you account for the energy used to increase the mass of water (from melting the conduit walls)? I could not understand the meaning of the phase transition temperature in this diagram - how is this expressed as an energy (yellow hash)? It seems to increase from the top to the bottom of the moulin (during pressurisation) and then to increase again on the passage to the margin (during depressurisation). This cannot be right as the change is in the opposite sense (the temperature decreases during pressurisation, and increases dur- ing depressurisation). Similarly, there is some confusion about gravitational potential energy, which should continue to decrease (becoming negative) in the moulin as the water goes below sea level, exactly cancelling out the 'overpressure'.**

Figure removed. At the advice of all reviewers, we have simplified the methods section and we

K. D. Mankoff                                                                                              p. 2 of 3

feel this figure is no longer needed.

**1  Specific Comments**

**Equation (1). Why use gamma here, and rho elsewhere, and in some places m. Couldn't the notation be made more consistent?**

This equation has been reemoved and the notation is now more consistent.

**Equation (3). What is z_o taken as here? Potential energy always needs to have a reference point, but it does not make sense to have a different reference point for different parcels of water (as seems to be the case here, since elevation at the terminus would vary) - comparison of potential energies is then meaningless.**

This equation has been removed.

**Equation (6). Delta phi here is a change in potential, but when plotted in the figures it seems to be a gradient (with units Pa/m). Which is it?**

Removed. But it is change in potential between one grid cell and the next, which is a gradient. We had used $\Delta$ as the discretized form of $\nabla$, but no longer do so.

**Figure 6. Why are there gaps in the data in panel c? There are also clear issues here with taking a numerical gradient of the discrete data, and some form of smoothing to calculate the gradient might yield cleaner results.**

Gaps are due to highly variable pressure gradient. There are cells in the model where water is routed across it, but the surface and bed of two adjacent cells are almost identical (flat) and therefore there is a minimal pressure gradient. This is now discussed explicitly in the text.

K. D. Mankoff

[git] ▪ TC @ f5ec86e [2016-09-01]

**Reply to Reviewer 2**

Ken Mankoff and Slawek Tulaczyk

Replies from the authors are inline in normal font and differentiated from the reviewer comments in bold colored font.

**Contents**

**1  General Comments**

**This work examines how specific climate change scenarios will increase the delivery of runoff-sourced gravitational potential energy to the bed of the Greenland Ice Sheet, which in turn will intensify formation of subglacial conduits there. The extent and size of conduits have (potentially profound) implications for ice velocity, which this manuscript touches upon but does not extensively discuss. Rather, the manuscript focuses on the assumptions and setup of the calculation. Its main conclusions are that gravita- tionally sourced heat dissipation exceeds geothermal flux and that conduit formation is expected to increase in future scenarios. These conclusions suggest the potential for slower ice velocity and enhanced sediment removal in land-terminating regions under future climate scenarios.**

**I found the idea behind this manuscript to be worthy and novel. However, the manuscript suffers from a lack of focus and is bottom-heavy, with a very involved description of assumptions (many of which are standard in subglacial hydrology) and the setup of the calculation, that builds to hesitant and limited larger arguments. These factors together limit the manuscript's publication readiness. I do think that the central idea is worthy and could be better highlighted, and its implications more thoroughly discussed, in future drafts.**

We are glad to hear the reviewer has found the idea worthy and novel. We have significantly re-written the methods section, and improved the results, discussion, and conclusions section in response to the reviewers suggestion. We note that re-arranging some of the results and discussion makes it appear as though more has changed than has actually been changed, when viewing the difference file.

[git] ▪ TC @ f5ec86e [2016-09-01]

**The manuscript is quite light on the implications of its calculation. In particular, it would benefit from a more thorough discussion of the effects of increased runoff on ice velocity; as it is now, this treatment is brief (Introduction only; it should be moved entirely to the Discussion) and greatly oversimplified. The authors state that increased runoff "appears" (P2 L6) to lead to overall slowdown, but this is based on only two cited stud- ies (Sundal et al., 2011 and Tedstone et al., 2015), both of which focus on beds under lower-elevation ice (s<1200 meters). The alternative (i.e., speedup) is not mentioned, even though this is an ongoing topic of discussion in the community. For instance, mul- tiple studies (e.g., Bartholomew et al., 2011 and Shannon et al., 2013) contrast with the cited studies, especially farther inland (s>1700 meters) from the ice margin (e.g., Doyle et al., 2014), where subglacial flow currently does not channelize and may not reach the "runoff threshold" required for channelization (Mayaud et al., 2014; cited in this manuscript). Because this manuscript considers subglacial conduit formation at these areas of the bed (s<2000 meters) as well as those near the margin (where the cited work is more applicable), it is crucial to recognize these distinctions and the cur- rent uncertainty in the runoff / speedup literature.**

The revised manuscript goes into much more depth regarding the implications of the calculations. We now make a clear distinction between marginal zone and interior ice, and the likely effects of VHD in these two distinct locations.

**With a more mature discussion related to ice velocity (or additional discussion of other im- plications hinted at; e.g., sediment evacuation), a reduction and better organization of the methods and assumptions sections, and attention to the points below (and those from AR1, which I largely agree with), I think this manuscript could make a good contribution**

As suggested we have a more in depth discussion, a reduction and better organization of the methods section, and have addressed all of the points raised below where applicable (the reor- ganization and removal of much of the methods section means a lot of the issues are solved by removal of the offending text).

**2   General Points**

**A good deal of the text in Section 3 (the details of each assumption) is unwarranted. These are standard assumptions that require a few citations, not multiple detailed para- graphs.**

We have rewritten the methods section entirely, and no longer discuss all the common assumptions that go into a subglacial hydrological model.

**PTT and energy conservation. It took me some time to work out why the authors claimed that this not zero-sum. My best understanding is that the PTT accounting begins at the bottom of the moulin (P5 L6), where the PTT is below zero. To cool to that temperature, the ice released sensible heat into the moulin (i.e., into the englacial, not subglacial, system). That sensible heat is, I think, part of the 35% of the gravitational potential energy that is lost (i.e., not transferred to basal ice). This needs to be made much clearer; perhaps with adaptations to Figure 1 (discussed later in this review).**

We have clarified that the model begins at the moulin bottom with a depressed PTT.

**By including the effects of the pressure-dependent melting point and using a relatively fine (5x5-km) grid, the authors obtain a detailed result of areas of expected basal freeze-on and of enhanced conduit formation. This has potential for application to local studies (and this**

K. D. Mankoff

**appears to have been tentatively investigated, e.g. comparison to Bell et al. (2014), although without great success), or for regional assessments (Northeast, Southwest, . . .) as commonly done in Greenland. Yet the authors discuss only ice-sheet-wide totals; I am not convinced that more than the back of an envelope was really needed for this.**

We have performed a more in-depth comparison with Bell et al. (2014) and discuss the regional implications of basal freeze on in more detail. In addition, we now highlight that the model freeze-on outputs are most likely indications where the routing model is inaccurate.

**I was distracted by the authors' use of numeric examples with multiple intermediate steps to demonstrate simple arguments. It would be simpler, more compact, and more elegant if the authors were to state the expressions they wish to compare, and then plug in the physical constants. For instance, P6 L18-20 is easily expressed as $PE/KE = 2\,g\,h/v^2 = 200$ for the stated $h$ and $v$. Another example is with PTT on P5 L2-5.**

We have removed the numeric examples.

**3  Specific Points**

**P1 L13-14 This sentence is difficult to parse**

We have clarified the phrasing of this sentence.

**P2 L6-9 This sentence is also unnecessarily complex, with multiple negatives, and contains two thoughts (descriptions of two studies) spliced together. Fundamentally, the logical flow is backwards, with the conclusion presented first and the evidence trailing behind.**

This sentence has been split and rearranged.

**P2 L13 data are**

Fixed.

**P2 L18 Mayaud et al. (2014) studied Paakitsoq, not Russell Glacier.**

Fixed.

**Regarding Mayaud et al. (2014): That study is a nice introduction to this manuscript and should also appear in the Discussion, not just the Introduction. That paper makes an explicit link between increased conduit formation and ice velocity that is only sum- marily discussed in this study; the authors could (and should) leverage this to make a more convincing argument about future ice velocity.**

We now discuss Mayaud et al. (2014) in more detail in the discussion.

**P2 L25-26 Pollard and DeConto (2012) – Similarly, this citation and future application belongs in the Discussion or Conclusion, not the Introduction.**

We have moved this text to the conclusion.

**P2 L28 extra word "a"**

Fixed.

**P3 L3 capitalize Glaciers**

[git] • TC @ f5ec86e [2016-09-01]

We prefer the lowercase "g". See `https://english.stackexchange.com/questions/199295/capitalization-rivers`

Regardless, I think this is an editorial decision for TC to make if this manuscript gets published.

**P3 L4 This section could use a short introduction that explains that all terms of the Bernoulli equation and all stages along the water's path will be examined to make the best approximation. The authors do part of this in P3 L14-17, which should be moved earlier in the section and emphasized.**

Equation removed.

**P3 L11-13 I found these two sentences to be a bit pedantic. At the very least, they are unnecessary because these terms are never again used in the manuscript.**

Removed.

**P3 L22 Figure 2 shows energy flux, not energy.**

Removed.

**P3 L24 Need to specify "in situ" for this statement to be correct.**

Fixed.

**P3-4 L28-3 The elevation z_s is set to a maximum of 2000 meters, even if the melt forms above there. However, in RCP8.5, runoff forms as high as 2800 meters (Figure 2d; South Greenland summit), which would contribute an additional 800 meters (+30%) of head. I would expect that when weighted with the small runoff volumes at high elevations, this addition would be small – but the effects of this approximation could be mentioned.**

$z_s$ is not set to a maximum of 2000 m, it is the actual GIS elevation (according to Morlighem et al. (2014)). Flow is routed to 2000 m before the subglacial model is initialized. This is the reason for the large step-change in initial volumes under the 4.5 and 8.5 scenario in the flowline figure. The ~800 additional m of head would only matter if the small amounts of water at 2800 m managed to access the bed at that location, which seems unlikely.

**P4 L8-9 This seems ancillary. I would encourage the authors to drop it or weave it in better.**

We think it is important to point out that not all model melt is considered "runoff".

**P4 L13 This reference is for alpine glaciers, which is fine, but this, and its applicability to Greenland, should be pointed out.**

This text removed during methods rewrite.

**P4 L18-19 The second clause is redundant with the preceding sentence.**

This text removed during methods rewrite.

**P4 L20 dissipates**

This text removed during methods rewrite.

**P4 L21 I would like to see some justification for why this heat is "lost", presumably to the atmosphere (as suggested in the legend of Figure 1). Does significant wind pumping really**

**occur at 50-100 meters depth inside a moulin? Alternatively, if this heat is "lost" to melting the moulin walls, this should be stated.**

This text removed during methods rewrite. Because we now start the model at the base under pressure, details of the transit from the surface to the bed are not needed.

**P5 L2 This is not an equation but an expression.**

This text removed during methods rewrite.

**P5 L7 This sentence suggests that this is not done for marine glaciers, although I believe it does need to be done (in an integrated-displacement sense, the water parcel does decompress to atmospheric pressure).**

If the marine terminus subglacial discharge rises to the surface it will reach atmospheric pressure. That seems to occur for most subglacial discharge, but that is outside the bounds of this model. We do not address processes at the vertical marine boundary, or under ice shelves, only grounded basal ice.

**P5 L18 The link to the Supplemental Material is broken.**

The supplemental material is available on the TCD website at `http://www.the-cryosphere-discuss.net/tc-2016-113/`

**P5 L24-30 The logical ordering here is flawed. The explanation for the 1/3, 2/3 values needs to be explained up front before the results are presented. As it is, the 1/3, 2/3 numbers appear mysteriously and add confusion until they are explained a few sentences later.**

This text removed during methods rewrite.

**P6 L 7-8 Choose either "downstream" or "down-stream" (I believe the former is more standard)**

Done.

**P6 L13 Strictly speaking and if you are willing to be arbitrarily precise, this is true of all water.**

True and we appreciate the precise wording. Unfortunately, this text removed during methods rewrite.

**P6 L25-27 Neither is this sentence true (due to hydrostatic pressure energy; see comment above for P5 L7) nor does it belong in this specific subsection.**

As stated, this is true for this model, which ends at the ice terminus. However, this text removed during methods rewrite.

**P7 L1-14 is way overblown. Furthermore, it reads like a response to a review, not like a journal article.**

This text removed during methods rewrite.

**P7 L18 Really do not need these details.**

This text removed during methods rewrite.

**P7 L19-20 It is unclear why this is relevant.**

This text removed during methods rewrite.

**P7 L27 "by the second law of thermodynamics" is completely unnecessary**

This text removed during methods rewrite.

**P8 L2 A "conjecture" is not a very convincing statement.**

This text removed during methods rewrite.

**P8 L5 "heating" here could refer to the water heating the ice, whereas I believe the authors are referring to sensible heating (PTT) of the water itself.**

This text removed during methods rewrite. But related to this, we are more explicit when discussing heating of ice versus change in internal heat content of the water itself.

**P8 L12-13 The figure does not show different units; both color bars show total annual energy flux. The only difference is /yr versus /sec. The "energy" unit stated in this sentence is incorrect.**

Fixed.

**P8 L30-31 It is unclear what error the instantaneous heat transfer assumption is intro- ducing. And "because due to" is a typo.**

Fixed.

**P9 L4-5 capitalize Russell and Leverett Glaciers; is estimated**

Fixed.

**P9 L8-9 The location of GHF = VHD appears to be coincident with the s=2000 meters location, the elevation where the authors first allow runoff to penetrate to the bed. This may be a coincidence; however, it does seem that the penetration contour would be a strong control on the amount of VHD at inland locations. How sensitive is this result to using s=1500 meters penetration contour, or a gradual ramp-up (i.e., 100% of available melt penetrating at s=1500 meters, 50% at s=1750 meters with the other 50% being routed downhill, and 0% penetrating / 100% routed downhill at s >= 2000 meters)? If the relative contribution of GHF versus VHD continues to be a primary conclusion of the study, this is a worthwhile exercise to include.**

The GHF = VHD line is not related to $z_s$ = 2000 m. That is the far right side of this graphic. $VHD_{8.5}$ = GHF occurs ~50 km margin-ward from that. $VHD_{4.5}$ = GHF occurs ~100 km (along-flow) margin-ward from the 2000 m contour. We do not see what coincidence the reviewer is referring to, but do agree that the penetration contour is a strong control. In fact, upstream of that contour, VHD should be effectively 0 (it is possible some small local routing feature would move water inland for a grid cell or two). Downstream, VHD increases as shown in panel c of this figure in a linear-log fashion.

**P9 L20-22 As far as I can reason, I believe this conclusion is backwards. If the authors' model included conduits, this would increase VHD near conduits, but at the expense of VHD at locations without conduits. Since conduits are more likely to form near the terminus than the interior, this inclusion would draw VHD away from the interior.**

This text removed from this section, but is mentioned at the end of the discussion when we address the impact of model resolution. We agree with the reviewers thoughts on margin/interior

location of increased heating. I think perhaps our imprecise language, where we did not specify what "near" or "far" meant with respect to conduit formation, may have led to the confusion.

**P9 L24 What specific components of flow routing and VHD are the authors referring to? Obviously, flow routing (a path) and VHD (a quantity) are not the same.**

This text removed during methods rewrite.

**P9 L28-29 Probably the authors mean that the DEM is not accurately representing the topography.**

Yes that too.

**P9 L31 How far, on average, are the Bell et al. (2014) basal freeze-on packages from where these results predict?**

We now address this in the text and include a figure.

**Regarding areas of basal freeze-on: This water will no longer follow subglacial hydropotential gradients, but instead surface gradients because it is attached to the ice. These are not always the same directions. This may or may not be important (likely it is not), depending on what fraction of water gets frozen on. This would be worth addressing briefly.**

Correct, but we are not tracking mass in this model, nor ice velocity. Nor does the model numerically address the latent heat released as the water freezes, although we do discuss it in the text.

**P10 L4 "conduits systems" typo; "non-conduit region" typo**

Removed.

**P10 L7-10 These sentences are awkward. Is it always the same grid cell? What are "such locations", probably marine-terminating glaciers with large catchments?**

Rephrased and clarified.

**P10 L11 These assumptions have already been stated in the Methods and Assumptions section, which is a more appropriate place for them than the Discussion.**

Removed.

**P10 L13 This should be phrased something like "the basally sourced meltwater carries away the initial gravitational potential energy of the runoff, in the form of latent heat, as it exits the subglacial system." The water itself is not heat.**

This text has been clarified.

**P10 L15-16 This sentence is interesting and important, and should be more prominent and/or pointed.**

This sentence is simply re-phrasing the fact that the bulk reported energy, 2.1 EJ, can melt 7 km$^3$ of ice. We think it is more informative to keep results in units of EJ. The additional water from melted ice is within the error bars of the total runoff (i.e. error bars are > 2%). This sentence has been removed.

**P10 L17-18 I do not think that any evidence for this statement has been presented. Also, "percentages" is vague – percentages of what?**

Text removed.

**P10 L21-22 According to Dow et al. (2014) (cited elsewhere by the authors), the sub- glacial conduit network is unlikely to expand significantly inland.**

Rephrased.

**P11 L2 Should specify "over land-terminating ice".**

Text has been removed.

**P11 L7 Should specify "total GHF integrated over the runoff area". As it reads now, it sounds like the solid earth will be warming in RCP 8.5.**

Clarified.

**P11 L10-13 This is a smorgasbord of facts. Needs better organization, logical flow, and build-up to the main idea.**

Rewritten.

**P11 L19 are P11 L21 cause**

Fixed.

**P11 L24-25 This idea is interesting, but the evidence the authors present suggests millennial timescales, not < 1 century.**

The change seems likely to begin on the century timescale, and may continue for millenia.

**P11 L27 missing "a"**

Added.

**P12 L2 missing "orders of [magnitude]"**

Added.

**P12 L6 Not just the ablation zone: the area with s<2000 m includes parts of the wet snow zone**

Fixed.

**P12 L8 Not "will become", as the authors have shown that it is already the dominant basal heat source (both P and H scenarios)**

Only near the margin. Clarified.

**P12 L8 "swamp" seems a bit informal P12 L28 its**

Fixed.

4    Figures

**Figure 1: The info is very small relative to the rest of the space, and the short definitions given to each colored box are inscrutable without reading the manuscript carefully. It is unclear where the water travels and why the bar can sometimes exceed 1. I had a very difficult time with it, and there are still components that I do not understand (what do the authors mean by**

[git] ▪ TC @ f5ec86e [2016-09-01]

"latent heat"? presumably that refers to VHD? why not call it VHD?). As a first figure, it is so inscrutable that it will drive away all but the most invested readers.

**Figure 1 presents results, whereas it is customary for the first figure to illustrate the setup. I think this figure would benefit from a major overhaul, with the results removed and a focused inclusion of methods components, such as the path of the water, the transfer of VHD into / out of the ice sheet along an undulating bed, input / output / conversion points for the various terms (gravitational PE, PTT), etc. I do not think anything would be lost from the manuscript if the colored bars were removed entirely.**

Figure removed. We feel that with the greatly simplified model description, this figure is no longer necessary.

**Figure 2: A negative sign is missing from the color bar label (W m^-2)**

Figure removed, as we are no longer referencing the $m\,g\,z$ potential energy source, and begin under the glacier with a pressure energy source.

**Figure 4: The units here are a bit confused. Delta-phi should be in Pa, as is phi. If the authors have divided by the 5-km distance between grid cells, that is Grad-phi and has the units shown on the color bar (Pa / m).**

Fixed.

**Figure 6: I agree with AR1 that the numerical gradient issues must be addressed.**

These are now addressed in the text. The variability in the pressure gradient is a product of the bed and surface DEMs, and may be real. Areas with low bed and surface slope will have low pressure gradient. We have smoothed the bottom graph (compared to the $\nabla\phi$ graph) to aid visual interpretation, and explain this in the figure caption.

**Panel a: see units (Pa versus Pa / m) comments on Figure 4**

Fixed.

**Panel b: Some of the lines show nonzero subglacial flow inland of the s=2000 meter contour (x 140 km), which the authors defined as the upper limit for melt to reach the bed. What is the reason for this?**

The surface does not cross the 2000 m elevation. I am not sure if the reviewer used an eye estimate, or something more precise. When I use a highlighting box in my PDF viewer I confirm that the elevation, and therefore the flow, does not cross 2000 m.

**Caption: should specify that b) is the flow rate of subglacial water**

Fixed.

**Figure 7 caption: use parallel definitions for the three cases; i.e., either all three are "the increase from A to B" or all three are "the difference between B and A". Also "Joules".**

Fixed. Also figure and caption changed to show results per basin, not per stream.

**Panel c: The symbol that the legend refers to is unclear.**

This figure has been regenerated and now shows each basin rather than the discharge from each stream.

**5    References**

Bell, R. E., K. J. Tinto, I. Das, M. Wolovick, W. Chu, T. T. Creyts, N. Frearson, A. Abdi, and J. D. Paden (2014). "Deformation, warming and softening of Greenland's ice by refreezing meltwater". *Nature Geoscience*. 7 (7), 497–502. DOI: 10.1038/NGEO2179.

Mayaud, J. R., A. F. Banwell, N. S. Arnold, and I. C. Willis (2014). "Modeling the response of subglacial drainage at Paakitsoq, West Greenland, to 21st century climate change". *Journal of Geophysical Research: Earth Surface*. 119 (12), 2619–2634. DOI: 10.1002/2014JF003271.

Morlighem, M., E. J. Rignot, J. Mouginot, H. Seroussi, and E. Larour (2014). "Deeply incised submarine glacial valleys beneath the Greenland ice sheet". *Nature Geoscience*. 7, 418–422. DOI: 10.1038/NGEO2167.

**Reply to Reviewer 3**

Ken Mankoff and Slawek Tulaczyk

Replies from the authors are inline in normal font and differentiated from the reviewer comments in bold colored font.

**Contents**

**In this manuscript, the authors calculated the viscous heat dissipation (VHD) generated as a result of runoff reaching the bed of the Greenland ice sheet, for the past and present, as well as for two future climate scenarios. The main findings are that VHD is becoming an increasingly large component of the basal heat budget – which is expected to contribute more significantly to subglacial conduits opening in the future.**

**I find the results novel and interesting, and a valuable addition to existing related work. However, the clarity of the text must be improved throughout, as the main or important points are often lost with too many details / repetitions / confusing sentences. Overall, I agree with comments aimed at clarifying the text, as given by AR1 and AR2. Below, I give a few more specific points below.**

We are glad to hear the reviewer finds the results novel, interesting, and valuable. In response to reviewer comments and suggestions we have clarified the text, simplified the methods section, and gone into more detail in the results and discussion sections. We note that re-arranging some of the results and discussion makes it appear as though more has changed than has actually been changed, when viewing the difference file.

**1 General Points**

**The discussion on the influence of subglacial hydrology and conduits formation on ice velocity (in particular under future scenarios), is over-simplified in the introduction and discussions (also pointed out by AR2). The overall effect of increased meltwater delivery to the bed of the ice sheet is unresolved. Some work suggest net deceleration (as already discussed), but other suggest a possible net acceleration (e.g., Bartholomew, NatGeo 2010; Doyle,GRL 2014). As the main implication from increased VHD concerns subglacial conduits formation, the authors should develop the potential implication of their findings more thoroughly.**

The discussion section now discusses the implications of our findings in more detail. In particular,

we now more clearly distinguish between the marginal zone and the interior, and the different effects increased VHD may have in these locations.

**2 Specific Points**

**P3 L12: define m**

Removed.

**P3 L24: likElihood**

Removed.

**P3 L27: suggest removing "(1-2 grid cells in our models)" – this is specified again later.**

Removed.

**P4 L29: Bring "these processes" to the same place in the text (water captured by crevasses and. . .?).**

Moved.

**P7: overall way too long, and hard to follow. What are the key points? P8 L13-14: Suggest replacing the sentence with a recall of Eq. 2.**

Methods rewritten. Removed.

**P8 L15: would write ". . .and 14.3 EJ year-1 (with 1EJ=1x1018J)"**

We have clarified the introduction of the EJ and (new) PJ notation.

**P8 L17-18: last sentence not necessary in my view.**

Removed.

**P8 L 30: remove "because"**

This sentence removed.

**P9 L25: sentence could be simplified – I find the use of statement such as " V times Eq (6)" clumsy.**

Removed.

**P9 L31: "perfect line up" between model and observations are rare, but it sounds like you were expecting it. It would be more useful to state how far apart the freeze-on packages are, and state where uncertainties might be coming from. Do you expect the errors from the model to relate to its physics, or input (GHF distribution, runoff distribution etc. . .)? . . . also, the advection argument seems far-fetched.**

We now discuss the distances between the two, and in more detail what we think basal freeze-on in our model means.

**P10 L11-13: very long sentence, the point is lost.**

Removed.

**P10 L14: numbered repeated from paragraph above. Suggest that section is reorganized to avoid repetition.**

These numbers are not a repetition. The first is the amount of subglacial discharge from one conduit. The second is the total GIS-wide volume of additional melt from VHD.

**P10 L32: Use EJ**

Removed, but care taken to use EJ or PJ elsewhere.

**P11 L1-2: This statement should at least be moderated, or could be removed, as this is an argument made (in a much more balanced way) in the conclusion.**

A major rewrite of the discussion better addresses the idea from this sentence.

**P11 L7: sentence describing the increase in GHF is not clear.**

Clarified.

**P12 L10: missing "and" after parenthesis**

This sentence has been rewritten in the revised manuscript.

3    Figures:

**Figure 1: Agree with AR1 and AR2, the bars and infos are vey small. Re-drawing with larger bars would help, as well as explicitly showing where VHD comes into the picture.**

Removed. Just as our methods was originally too detailed, we do not think this figure is needed when discussing the standard assumptions of subgalcial hydrological theory.

**Figures 2 -3 -5: Not sure if there would be space for this, but I feel like these would beneficiate from being enlarged, e.g., as a 2 lines / 2 columns panels presentation. This is particularly true for Figure 5, where it is very hard to see any freeze-on.**

2x2 panels are now used. Freeze-on is hard to see in this figure, but the new figure comparing with Bell (2014) should be clearer.

[revised manuscript text omitted]

The majority of studies Studies examining surface melt, supra-glacial routing, subglacial hydrology, and the response of ice sheet outlet glaciers to those various inputs takes place take place predominantly in southwest Greenland, often near the Russelland/or LeverettGlaciers focusing largely on the Russell, Leverett, Paakitsoq, or nearby glaciers
130 (for example, \xdef 10001000 \xdef 10001000 Banwell:2013Modelling,Arnold:2014High-resolution,Andrews:2014Direct,Tedstone:2015Decadal). Furthermore, present day weather, runoff, outflow, and other data is are often used in those studies, since daily, hourly, or higher temporal resolution of the data is beneficial to the models. However, using present data limits their focus to some this approach limits the focus of these studies to
135 recent seasons for which abundant in-situ sensor data exists exist. In order to examine future scenarios, \xdef 10001000 \xdef 10001000 Mayaud:2014Modeling built on the work of \xdef 10001000 \xdef 10001000 Banwell:2013Modelling, but used a conduit model that includes melt opening and creep closure, driven by a positive degree day runoff model, to examine future changes to year 2095 under various IPCC RCP scenarios
140 Moss et al. 2010, (Moss *et al.*, 2010). Those models had hourly or daily resolution and were again limited to the well-studied ~200 km$^2$ area near Russell glacier southwest sector.

Here we perform a broader analysis that examines uses runoff over the entire GIS on annual and decade timescales, and frame the discussion in terms of changes in heat
145 available to melt open conduits (VHD) available heat, rather than focusing on water pressure in a conduit relative to overhead ice pressure. We report both the total GIS-

wide energy budgets, its distribution   and its distribution per basin and   at a 5x5 km grid resolution, and highlight results along one flowline at   . We also highlight a high-resolution (  150 m resolution. We examine what fraction of the initial gravitational potential energy of surface meltwater converts to heat subglacially and is used for subglacial melting. Our treatment of en- and sub- glacial hydrology is simplified because it does not represent actual conduits, but is at the same time more comprehensive than in existing global climate or ice-sheet models (e.g. Pollard and DeConto (2012)) , and may offer a computationally efficient yet improved method to incorporate glacial hydrology at their existing grid resolution  m) calculation near Petermann Glacier and along a single ice flowline in southwest Greenland  .

**3   Data**

We use a   150 m resolution basal topography and surface topography (IceBridge BedMachine Greenland, Version 2) from **\xdef** 10001000 **\xdef** 10001000 Morlighem:2014Deeply,Morlighem:20 to calculate both surface and subglacial flow routing and subglacial pressures.  Surface runoff, equal to the surface meltwater and rain that does not re-freeze but instead runs off  plus rain less refrozen water  , comes from MARv3  MAR v3  .5.2 Fettweis et al. 2013, (Fettweis *et al.*, 2013)  .

We report results for a historical period (1900  1960  – 1999), the present (2010 – 2019), and IPCC AR5 RCPs 4.5, and 8.5 (2090 – 2099) . We  Moss et al. 2010, (Moss *et al.*, 2010). We also highlight a baseline (TB, 1985–1994) and reference (TR, 2007–2014) period that match the baseline and reference periods in Tedstone et al. (2015, Tedstone *et al.* (2015)). It is important to note that while our TB may match the baseline data from Tedstone et al. (2015, Tedstone *et al.* (2015)), there is no expectation that TR matches the reference data from Tedstone et al. (2015, Tedstone *et al.* (2015)). This is because the TB time frame use ERA-Interim reanalysis products, but the TR time-frame is from a "future" projection simulation Fettweis et al. 2013, (Fettweis *et al.*, 2013), and is unlikely to have simulated the specific annual runoff, including the extreme melt in 2012.

We   process the entire GIS at 5 km resolution, and then   the area near Petermann Glacier at 150 m resolution, and   part of West Greenland (near the Russell and Leverett glaciers) and   at 150 m resolution, where we extract a   the sample   flowline segment.

**4   Methods and assumptions  Model Description**

We introduce   use a flow-routing and energy balance model that incorporates common assumptions about glacier hydrology (e.g. Röthlisberger (1972, Röthlisberger (1972)), Shreve (1972, Shreve (1972))), but does not explicitly resolve subglacial conduits. We lay out   our methods and assumptions by tracing the path of a unit parcel of surface meltwater from source to sink (ice surface elevation at the origin of a meltwater parcel to sea or land-outlet level   submarine or terrestrial outlet   where it discharges from an   the ice sheet). We follow a simplified form of Bernoulli's equation,

$$\frac{p_s}{\gamma} + z_s + \frac{v_s^2}{2g} = \frac{p_b}{\gamma} + z_b + \frac{v_b^2}{2g} + H_L,$$

with $p$ pressure, $\gamma$ the specific weight of water, $z$ elevation above sea level, $v$ velocity, $g$ gravitational acceleration, and $H_L$ combined head loss (e.g. Gulley et al. (2014)). Subscripts $s$ represent the ice surface and $b$, the bed. Viscous heat dissipation (VHD) is the process that causes head losses $H_L$. Eq. 4 is in units m and the three terms are often referred to as the pressure head, elevation head, and velocity head. It can be multiplied by $m\,g$ and then it is an energy balance equation, with the middle term the gravitational potential energy.

We describe the model used to route surface runoff  We assume all surface runoff that begins at elevations above 2000 m is unable to leave the surface or penetrate  to the bed and down the hydropotential gradient, where subglacial conduits melt open from VHD in flowing water (Fig. **??**). At each step we detail the assumptions made, the amount of energy available to the parcel of water with respect to the initial gravitational potential energy, and the form of that energy: gravitational potential, kinetic (velocity), pressure, or transferred out of the water parcel as sensible heat.

**4.1  Surface runoff and routing**

Surface runoff comes from melted ice or rainfall that does not refreeze or evaporate. At its source it has a total gravitational potential energy, $PE_{\text{total}}$,

$$PE_{\text{total}} = m\,g\,z_s.$$

Results of Eq. (4.1) are shown in Fig. **??**. We assume that initially water has only its gravitational potential energy available and has negligible velocity and kinetic energy and it is initially at 0 °C at atmospheric pressure.

Runoff with an initial elevation > 2000 m has low liklihood of penetrating to the bed  , and is instead routed  at those elevations Poinar et al. 2015, (Poinar *et al.*, 2015). Instead we route it  on the surface to below  the  2000 m elevation . Elsewhere, runoff may leave the surface within its source grid cell. Side-draining lakes and supraglacial streams transport horizontally by definition, and moulins often drain these to the bed near their source  . Surface meltwater may on average flow  contour. All surface runoff at or below 2000 m elevation is assumed to access the bed within the  5 – 10 km (1 – 2 grid cells in our model)  km square grid cell in which it originates Yang and Smith 2013, (Yang & Smith, 2013). In reality water may flow slightly farther before leaving the surface . However,  Yang et al. 2015, (Yang *et al.*, 2015), but  we ignore this horizontal transport because when streams do travel this far on the surface, they are most likely to do so in gently-sloped and  crevasse-free regions of ice sheets that are subjected to relatively low strain rates and stresses  implying a flatter surface Poinar et al. 2015, (Poinar *et al.*, 2015) . Horizontal transport of surface meltwater with

a small elevation drop has implies only a small impact on the initial gravitational potential energy. Horizontal calculated VHD. Horizontal surface transport in an area with large surface slope and impact on gravitational potential energy slopes is unlikely, because such regions have high driving stress, and crevasses routing the stream to the bed are likely to be present.

Near-surface water Water storage may occur , but because we discuss results on an annual and longer timescale and focus on change over time, only changing relative amounts of storage would affect the results, and only in amounts larger than in the firn or in crevasses. We assume these volumes are insignificant (at most a few percent of ) relative to the total runoff volume. Near surface water storage and refreezing are not addressed here but the surface mass balance model that generates the runoff takes some refreezing into account . amount, that percent is not likely to change much in the future, and englacial storage does not release heat at the bed, which is the focus of this study.

**4.1 Transit from the surface to the bed**

Once at the bed, flow routing moves water in the direction of the negative of the gradient of the hydropotential $\phi$ Shreve 1972, (Shreve, 1972),

We assume all water reaches the bed within the 5x5 km grid cell where it leaves the surface. Moulins and crevasses, either at a lake bottom or when a supraglacial stream leaves the surface, carry water from the surface to the bed. Although the exact path to the bed is poorly constrained, moulins generally deliver their water to the bed within a few ice thicknesses .

$$\nabla \phi = \rho_w \, g \, \nabla z_b + \alpha \, \rho_i \, g \, (\nabla z_s - \nabla z_b), \tag{1}$$

When falling through the air-filled portion of a moulin or crevasse, part of the initial gravitational potential is converted to kinetic energy and then dissipated as heat to the surrounding air, moulin and crevasse walls, and water table surface on impact.

We assume the density of ice is 917 kg m$^{-3}$, with $\phi$ the hydropotential (units Pa), $\rho_w$ the density of water is meltwater ( 1000 kg m$^{-3}$, and that the subglacial system is pressurized to slightly less than the ice-overburden pressure as often observed in the field (Engelhardt and Kamb (1997), Fountain (1994), and Meierbachtol et al. (2013)). Given the above, the water table in the moulin will be ~90% of the way up the moulin when the subglacial system is near the ice-overburden pressure. In this case, stationary water at the surface of a 90% water-filled moulin retains ~90% of its initial gravitational potential energy and dissipated ~10% to heat (Fig. **??**) , which we consider "lost" from the system. The potential energy available at the moulin water table surface relative to the terminus elevation is therefore a subset of $PE_{\text{total}}$ (Eq. 4.1), equal to,

$$PE_{\text{moulin}} = 0.9 \, m \, g \, (z_s - z_b) + m \, g \, (z_b - z_o),$$

K. D. Mankoff

with $z_o$ the elevation of the outflow at the terminus. Moulins do not conduct significant
heat to ice , and while we assume all potential energy is converted to pressure energy
along the descent, we also assume that water remains at the phase transition temperature
(PTT),

$$PTT = C_T \left( z_s - z_b \right) \rho_i g$$

derived from $C_T$ the Clausius-Clapeyron slope equal to 8.6x10$^{-8}$ K Pa$^{-1}$  , and  ), $g$
gravity, $z_b$ the bed elevation, $\alpha$ a flotation fraction (set to 0.9) because the subglacial
system is often slightly less than the ice-overburden pressure (Engelhardt and Kamb
(1997), Fountain (1994), and Meierbachtol et al. (2013, Engelhardt & Kamb (1997), Fountain
*et al.* (1994), Meierbachtol *et al.* (2013))),  $\rho_i$ the density of ice . Water captured by crevasses
also warms ice as it refreezes . We do not consider either of these processes that release
heat englacially because they do not occur at the bed, and the crevasse-captured water
volume is negligible relative to the total surface runoff volume.

In our model, the PTT adjusts the total energy by ~30% and is not zero sum even though
input and output are both at atmospheric pressure (PTT = 0 °C). From the specific heat of
water ($c_p$ = 4190 J kg$^{-1}$ K$^{-1}$  (917 kg m$^{-3}$ ), and the equation $(g\,z)/c_p$, water can warm
0.0023 °C per m of elevation drop. A unit mass of water at  $z_s$ = 1000 m has gravitational
potential energy of ~10000 J and can therefore warm itself by 2.3 °C if lowered to 0 m.
From Eq. **??**, the Clausius-Clapeyron slope reduces the PTT -0.7 °C due to the pressure
under 1000 m of ice, equal to ~1/3 of the temperature change from gravitational potential
energy. However, as stated previously we do not consider this initial change in PTT in the
moulin because that occurs englacially. As the water flows out and returns to atmospheric
pressure (for land-terminating glaciers), we do consider changes in the PTT, meaning
~1/3 of the potential energy may used internally by the water to increase its temperature
with the PTT as the PTT returns to 0 °C  the surface elevation .

**4.1  Flow routing at the bed**

Once at the bed, water is routed from one grid cell to the next based on the gradient of
the hydropotential $\phi$ calculated from the ice surface and bed elevation, and decomposed
into elevation potential $\phi_z$, and pressure potential $\phi_p$,

$$\phi = \phi_z + \phi_p = \rho_w\, g\, z_b + \rho_i\, g \left( z_s - z_b \right),$$

with $\phi$, $\phi_z$, and $\phi_p$ in units Pa. $\phi$ equals the potential energy per unit volume of water,
and dividing by $\rho_w\, g$ gives the hydraulic head  .   All water is assumed to move
to the neighboring cell (of 8)  one neighboring cell  with the lowest hydropotential
and conduits are not included in the model. This flow routing redistributes the surface
source into subglacial streams (Fig. 12.1).   with eight total neighbors considered.
Flow routing is implemented using the `r.watershed` tool in grass GIS  GRASS GIS

Neteler et al. 2012, (Neteler *et al.*, 2012) version 7.0. .4 , with $\phi$ as the "elevation" input with all  local minima filled so that all water leaves the ice sheet (see Supplemental Material).

The change in hydropotential that drives flow comes from a combination total hydropotential can be decomposed into an elevation term and a pressure term, where the former is the first term on the right hand side of Eq. (1 ) and the downstream cell from the flow routing algorithm, or,

$$\Delta\phi = \phi_i - \phi_{i+1},$$

where water flows from grid cell $i$ to cell $i+1$. $\Delta\phi_z$ 1 and $\Delta\phi_p$ are defined similarly to Eq. (**??**).

The net hydropotential change $\Delta\phi$, does not distinguish between the latter is the second term. We use this decomposition to examine in more detail the spatial distribution of flow driven by changes in basal elevation which does not change the PTT the bed elevation , and flow driven by changes in ice thickness or pressure which does change the PTT. Our decomposition of $\Delta\phi$ into $\Delta\phi_z$ and $\Delta\phi_p$ (Eq. 1 and **??**, Fig. 2) allows us to estimate where potential energy losses occur due to an elevation drop and where potential energy losses occur due to a pressure drop. This distinction matters because if flow is driven by an elevation drop (red $\Delta\phi_z$ in Fig. 2), ice thickness may increase, decrease, or remain constant, and those three scenario imply a $\pm 2/3$ change in the pressure gradient.

The water remains at the pressure-dependent phase transition temperature (PTT) and energy is released based on the change in hydropotential combined with the changing phase change temperature. Because our focus is on energy available at the glacier bed, we ignore heat released due to changing PTT down the moulin (e.g. Catania and Neumann (2010, Catania & Neumann (2010))), and the model is initialized at the moulin bottom with a depressed PTT.

Our energy budget model tracks energy between inputs at the ice sheet bed where energy begins as either pressure or gravitational potential energy (which may be net positive if the source bed elevation is above the discharge elevation, or negative if it is below), and the pressure change.Similarly, if flow is driven by a pressure drop (red $\Delta\phi_p$ in Fig. 2), ice must thin along-flow, and ~ output where the energy is in one of three forms: 1/3 of the potential heating from the pressure drop is not released . These ~1/3 and ~ ) the latent heat of cumulative basal melt caused by VHD released along the subglacial water flow pathway, 2/ ) gravitational potential energy of discharge from land-terminating glaciers with terminii above seal level, or 3values come from the PTT, the specific heat of water, and the density of water, or $C_T c_p \rho \approx 1/3$ ) pressure if discharged below sea level from a marine terminating glacier .

**4.1   Basal viscous heat dissipation**

325 Between the input and discharge locations, all energy is assumed to dissipate as heat to the surrounding ice within the grid cell where the energy transfer occurs Isenko et al. 2005, (Isenko *et al.*, 20 As water flows down the hydropotential gradient, we track the energy released as heat( , Q) , based on the volume of water, the change in the hydropotential, and the change in the PTT,

$$Q = V(\underline{\Delta\nabla}\phi - C_T\, c_p\, \underline{\Delta\nabla}\phi_p\, \rho_w) \tag{2}$$

330 where $V$ is the volume of water (Fig. 12.1), $\Delta\phi$ is the net hydropotential change along flow from cell $i$ into $i+1$, and $\Delta\phi_p$ is the    $\nabla\phi$ the hydropotential gradient, $C_T$ the Clausius-Clapeyron slope equal to $8.6\mathrm{x}10^{-8}$ K Pa$^{-1}$ Hooke 2005, (Hooke, 2005), $c_p$ the specific heat of water equal to 4184 J K$^{-1}$ kg$^{-1}$, $\nabla\phi_p$ the    pressure component of the hydropotential change along flow  gradient, and $\rho_w$ the density of water  . The last 335 term of Eq. (2) is an   the   adjustment for the PTT, which increases the heat released along the flowline when the ice thickens down-stream ($\Delta\phi_p$ is negative,   downstream and the PTT drops) , and reduces the heat released along the flowline when the ice thins downstream. If the    second term on the RHS    right hand side   of Eq. 2 is larger than the first termon the RHS  , then $Q$ is negative, and basal freeze-on occurs 340 Alley et al. 1998; Bell et al. 2014, (Alley *et al.*, 1998, Bell *et al.*, 2014)

**5   Results**

**5.1   Energy advected from the system as sensible heat or velocity  Volume of subglacial hydrology**

In an idealized scenario where water exits a glacier at sea level, it will have lost 10% of its 345 initial gravitational potential energy at the beginning of its journey through the air-filled portion of the moulin and dissipated the remaining 90% as heat during its flow through the englacial and subglacial drainage system. In reality, most water leaves glaciers above or below sea level. When water leaves the ice sheet margin from a land-terminating glacier all water pressure in excess of atmospheric pressure is released and only gravitational 350 potential energy corresponding to the elevation of the glacial outlet above sea level remains   Subglacial hydrology flow volumes are a combination of the surface runoff and the flow routing algorithm. Annual average runoff volume has historically been 244 km$^3$, presently is 345 km$^3$, and in the future will be 524 or 1278 km$^3$ under the 4.5 or 8.5 scenarios. Because we do not account for changes in water storage, total subglacial 355 volumes are the same. The spatial distribution of flow volume at the bed matches the large-scale surface distribution - more occurs in the south than the north, and the bulk occurs in the southwest sector. Most runoff also occurs at the edge of the ice sheet. Under RCP 8.5 it is predicted to occur at all elevations in south Greenland, but here only accesses the bed when the surface elevation is < 2000 m   (Fig. **??** 12.1 ).

360 Only a small fraction of the total gravitational energy drop is converted to kinetic energy (velocity)   . This can be illustrated by the fact that a 1 kg unit parcel of water has

K. D. Mankoff

gravitational potential energy of  Flow routing of basal hydrology causes orders of magnitude difference in water volume over small spatial distances as streams collect and discharge the water. At present, the largest volume of discharge is  ~10000 J when it is at 1000 m elevation, and kinetic energy of ~50 J (only 0.5  7 km$^3$ year$^{-1}$ from a single grid cell (2  % of the initial gravitational potential energy)if it flows out of the terminus at a relatively fast velocity of  total annual runoff - that percentage has not and does not significantly change). The volume flow rate has increased from ~5 km$^3$ year$^{-1}$ historically, and is  10 m s  or 31 km$^3$ year $^{-1}$ under the future 4.5 or 8.5 scenarios.

**5.2  Pressure v. elevation driven flow**

The zone around Greenland with active subglacial flow has distinct regions where the flow is driven by changes in elevation (Fig. 2 panel $\nabla\phi_z$) or pressure (Fig. 2 panel $\nabla\phi_p$) . Given that meltwater is at atmospheric pressure and in contact with ice between the entrance (e. g., moulin) and exit (e. g., ice-marginal tunnel)there should be no significant net change in water temperature and no significant change in heat content between the input and output ends of the hydrological system. Hence, the bulk of the gravitational energy loss experienced by surface meltwater will be dissipated as heat during subglacial water flow  Flow always leaves the ice sheet at the margin due to pressure-driven flow (i.e. from regions with thinning ice, red outer band in Fig. 2), but inland often travels under regions where ice thickens and pressure increases (blue regions in Fig. 2). Distinct regions of flow under thickening ice occur near the Petermann, Zachariae Isstrøm and 79 North Glaciers, and some coherent patches along the west coast . Early work in glacier hydrology indicate that this heat is used to melt icein contact with the flowing water . When water leaves an ice margin under a marine-terminating glacier, the same processes occur releasing heat subglacially, but the water at the exit point does not return to atmospheric pressure but rather to the pressure determined by the depth below sea level at which subglacial outflow takes place (Fig. **??**).

Water also does not leave the system with significant energy in the form of sensible heat. Two separate arguments support this proposition, in addition to the bulk of existing subglacial fluid thermal transfer literature:  Large differences in released heat (>35%) are due to flow under thinning or thickening ice. When water flows under thinning ice, ~35% of the heat released by the reduction in pressure is used to warm the water with the rising PTT Röthlisberger 1972, (Röthlisberger, 1972), and not included in our VHD numbers. In the regions highlighted above where flow occurs under thickening ice, a decrease in the PTT increases the VHD term.

**5.3  Flow-routed spatial distribution of VHD**

A spatial map of basal VHD is shown in Fig. 3 with the energy calculated based on Eq. (2). Summing the spatial data in Fig. 3 gives annual GIS-wide VHD of 1.4, 2.1, 3.5, and 9.8 EJ year$^{-1}$ ( 1 ) laboratory experiments, and 2) measurements in proglacial streams EJ = 1x10$^{18}$ J) for the historical, present, RCP4.5 and RCP8.5 cases respectively (Table 1) .

The standard assumption in the subglacial hydrology literature is instantaneous heat transfer , combined with all heat being delivered to the ice or used to maintain the water at the phase-change temperature , with only the following few exceptions: Mathews (1973) examined subglacial outflow from an active volcanic/geothermal terrain where water may be starting with temperatures far above the freezing point. Hock and Hooke (1993), when re-analysed with the correct heat capacity of water (pers. comm.; the heat capacity of water is not 256.9 J kg$^{-1}$ K$^{-1}$ as in Hooke (2005)) report that 15% is advected. Röthlisberger (1972) assumes instantaneous heat transfer for the bulk of that paper, but on page 197 suggests At present a maximum up to ~1 W m$^{-2}$ is released where the largest volumes of water leave the ice sheet, over an entire 5x5 km grid cell. More generally, between 0.1 and 1 W m$^{-2}$ is released in the marginal zone, but by year 2100 under RCP8.5, this amount of heat is likely to be released throughout almost the entire area of GIS where runoff is projected to reach the bed. In the future, regions with high discharge may experience 10 - 50% of energy may leave a subglacial system unused. However, his citation is personal communication with Mathews (in press), cited as W m$^{-2}$ VHD rates.

In some regions, heating is "Mathews, W. H. In press. Record of two jökullhlaups (1969) negative ". We find no such paper, but Mathews (1973) has an identical title and reports through one method that only 20-50% of energy is used to melt ice (50-80% leaves the system as sensible heat), which indicates basal freeze-on. These regions are a subset of the regions where pressure increases due to ice thickening along-flow ($\nabla\phi_p$ in Fig. 2). Locations of basal freeze-on occur throughout the GIS, including near Petermann where Bell et al. (2014, Bell *et al.* (2014)) provides observational evidence of packages of basal freeze-on. Our model estimated locations (blue in Figure 7), and through another that 10 - 50% is lost to advection (as quoted by Röthlisberger) . A manuscript written by Rist (1954) in Icelandic is interpreted (translated) by two different sources. Nye (1976) reports 0.05 °C (water can be warmed 0.05 °C by lowering it 21 m) but approximates that as 0 °C, and Björnsson (2010) reports that Rist (1954) has repeatedly measured outflow at 0 °C. We are aware of no other manuscripts suggesting that sensible heat is advected from the subglacial environment with discharging water and the few examples to the contrary come from either glacial systems in Icelandic volcanic terrain (e.g. Mathews (1973))or a glacier with a very short drainage pathway .

Laboratory experiments indicate that water flowing through ice reaches a near-zero equilibrium temperature within 10s to 100s of m . An independent method measuring and modeling proglacial stream temperature closes the heat budget with an outflow of 0 °C, with stream temperatures above that attributed to one of four sources occurring between the glacier snout and thermometer: net shortwave radiation, evaporative heat flux, sensible heat flux, and streambed friction . The lab and pro-glacial stream experiments do not address temperature changes due to the pressure-dependence of the phase change of water.

Even if all heat is consumed under the glacier, the assumption that all heat is transferred to the ice is likely violated as some energy may be used for eroding and/or transporting debris, fracturing ice, generating seismic waves, and to heat subglacial materials .

K. D. Mankoff

445 We assume that these terms are negligible, and note that some of these, like heating of subglacial materials, are likely to be intermediate processes that themselves will re-release heat. We therefore neglect these terms, assuming as much of the existing subglacial hydrological heat transfer literature does that melting of basal ice represents the ultimate sink of heat beneath wet-based ice masses. Due to background geothermal
450 flux, subglacial geologic materials will normally be warmer than ice and , by the second law of thermodynamics, heat will flow from the warmer subglacial material towards colder ice.

We treat the energy used to transport bedload as net zero within each model grid cell. Because the energy used to pick up bedload is returned when the bedload drops, only
455 bedload transported across the terminus impacts the total energy available for heating. Large volumes of fine sediment are carried across the terminus boundary, but the kinetic energy of that sediment is a small fraction of the kinetic energy of the water (which was previously shown to be negligible), and we therefore do not consider it Bell et al. (2014, Bell *et al.* (2014)) observed locations (black in Figure 7) show some agreement and some
460 disagreement. We interpret the disagreements as a combination of artifacts in the basal DEM and artifacts due to limitations in our routing model. We address each of these in the discussion section .

**5.4 Methods and assumptions summary Basin-scale changes of VHD**

We conjecture that nearly all the gravitational energy loss experienced by surface melt-
465 water as it travels through and beneath the Greenland ice sheet is ultimately used to melt ice in contact with englacial and subglacial water drainage pathways. Our model closes the energy budget by assuming that 90%of the potential energy loss experienced by surface meltwater is simply balanced by heating and melting under the ice sheet in the ideal scenario of a flat bed and outflow at sea level. This is equivalent to stating that all
470 terms on the RHS of Eq. 4 approach 0 at the outflow, except for $H_L$, which must balance the high elevation term on the LHS. If some fraction of the gravitational energy loss of water flowing under ice sheet is converted into some other form of energy neglected here, it is worth noting that the central focus of this paper is on examining changes over time, and the impact of the neglected terms is muted as long as they do not change
475 relative magnitude over time Basin-scale changes between the three different time periods considered here are well illustrated when viewed as change (units Joule) in VHD per basin (Fig. 4) or percent increase (units %) in VHD per basin (Fig. 5). Basin size influences results for the former, and the effect is removed for the latter. Because integrated per basin VHD removes the effect of flow routing, VHD per basin is approximately proportional to
480 runoff per basin, and changes in basin VHD are proportional to changes in basin runoff .

**6 Results**

**5.1 Gravitational potential energy**

The source term, gravitational potential energy of surface runoff, is shown in Fig. **??** in units of both energy and as annually averaged energy flux rate (i.e. power) . Energy measured in Joules comes from the runoff volume multiplied by density to get mass, and power in Watts comes from dividing the energy by the number of seconds in a year Change between the Tedstone et al. (2015, Tedstone *et al.* (2015)) reference period (TR) and the Tedstone et al. (2015, Tedstone *et al.* (2015)) baseline period (TB, Fig . Summing across the ice sheet 4a) shows a 2 PJ year$^{-1}$ (1 PJ = 1x10$^{15}$ J) increase in the energy in each basin in the southwest sector where Tedstone et al. (2015, Tedstone *et al.* (2015)) observed a general velocity decrease. (A 2 PJ year$^{-1}$ increase means a cumulative increase of 2 PJ in the future years, not a rate of change of 2 PJ each year between the two time periods). Elsewhere, increases were minimal (southeast) or negative. A similar pattern emerges between the historical and present cases (Fig. **??**)gives annual GIS-wide gravitational potential energy of 2.1, 2.9, 4.9, and 14.3 times 4b), with the bulk of the change in the southwest sector, but larger than for TB-TR. At present there is a 10$^{18}$ J (or exajoules, EJ) PJ year$^{-1}$ for the historical , present , RCP4.5 and RCP8.5 cases respectively after the surface runoff is routed to the 2000 m contour (Table 1). The same result can be found by evaluating Eq. (4.1)with total annual runoff volume of 345 km$^3$ times 1000 kg m$^{-1}$ and difference compared to the historical rate. Between the present and the volume-weighted mean elevation of 863 m for the present day case (Table 1) 2090s under the RCP 4.5 scenario, 10 PJ year$^{-1}$ increases occur in all sectors except the northwest (Fig. 4c). In the RCP 8.5 scenario, 100 PJ year$^{-1}$ increases occur in several basins (Fig. 4) .

**5.1 Viscous heat dissipation**

Percent increase between historical and present shows that the increases scale with latitude. All of Greenland has experienced an increase, with many regions showing a 2- to 3-fold increase (+100-200%) in VHD (Fig. 5 panel P/H). Runoff, and therefore VHD, in the north of Greenland has experienced the largest percent increase. This is because VHD values are so small there that all increases appear large when viewed on a percentage scale.

A spatial map of basal VHD is shown in Fig. 3 with the energy calculated based on Eq. (2). Summing the spatial data in Fig. 3 gives annual GIS-wide VHD of 1.4, 1.9, 3.2, and 9.0 EJ year$^{-1}$ for the historical, present , RCP4.5 and RCP8.5 cases respectively(Table 1) . These numbers are However, the choice of baseline matters. The historical and present periods are 1960 – 1999 and 2010 – 2019 respectively. If the TR (1985 – 1994) and TB (2007 – 2014) periods are used instead, our results instead match the results from Tedstone et al. (2015, Tedstone *et al.* (2015)), which showed a 50% increase in runoff in the basins just south of Jakobshavn Isbræ. The issue with our TR data produced by a future simulation and not using reanalysis products may be the cause of the difference between a ~65% of the incoming gravitational potential energy respectively, which is below the theoretical value of ~90% discussed above. The difference is due to initial bed topography below sea level, land terminating glaciers discharging above sea level, and 50% increase in runoff reported by Tedstone et al. (2015, Tedstone *et al.* (2015)) and the pressure dependence of the phase transition temperature. For example, if a parcel of water begins at 1000 m

K. D. Mankoff                                                                                    p. 14 of 33

elevation over a -1000 m bed and discharges at 100 m elevation (Fig. **??**), the following processes reduce VHD: 1) the 90% of ice overburden basal pressure means that the moulin water level is at 1800 m above the bed (800 m above sea level), and 20% instead of 10% of the initial gravitational energy relative to sea level is dissipated before surface meltwater reaches the water table (Eq. 4.1), 2) water discharges at 100 m elevation rather than at sea level, so the total potential energy drop through the englacial and subglacial drainage system is due to 700 m of elevation change (Eq. 4.1), and ~10-20% increase in VHD obtained here.

**5.1   VHD along a flowline in southwest Greenland**

Viewing results along a flowline highlights that the hydropotential gradient driving the flow becomes spatially both larger and more variable toward the margin (Fig. 6a). Along a single flow-line, step-increases in volume occur where other major tributaries join the tributary displayed here, causing 3 to 4 orders of magnitude increase in flow volume (Fig. 6b). This increase in water volume leads to a   3 ) the impact of   to 4 orders of magnitude increase in VHD (Fig. 6c). Variations in bed topography and ice thickness create variations in the gradient of $\phi$ along the flowline (Fig. 6a) and therefore variations in VHD along the flowline (Fig. 6c). Although the general trend of VHD increases from inland to the margin (Fig. 6c) due to increasing flux (Fig. 6b),   the PTT should be net 0 from inflow to outflow because both are at atmospheric pressure, but is not net 0 because due to the instantaneous heat transfer assumption, and the fact that our model only tracks VHD at the bed (Eq. **??**). Water that enters the model at the moulin bottom is at the PTT below 0 °C, and energy is consumed warming it as the ice thins (Eq. 2)   hydropotential gradient (Fig. 6a) adds a high variability signal to the background flux-driven signal, with 1-2 orders of magnitude change in VHD over just a few of the 150 m grid cells. Gaps in Fig. 6c are due to low gradients at those locations causing the release of only minor amounts of VHD  . Beyond this specific example, if outflow occurs at marine terminating glaciers under thicker ice than where the water entered the system, that water has the ability to release excess heat due to the PTT.

**5.2   VHD and GHF**

Frictional basal heating is up to 0.2 W m$^{-2}$ near the Russell and Leverett glaciers Brinkerhoff et al. 2011, (Brinkerhoff *et al.*, 2011)  , while geothermal heat flux (GHF) is estimate   estimated   at ~0.050 W m$^{-2}$ Shapiro and Ritzwoller 2004, (Shapiro & Ritzwoller, 2004) to as little as 0.030 W m$^{-2}$ Meierbachtol et al. 2015, (Meierbachtol *et al.*, 2015)  . The logarithmic scale used in Fig. 6c makes the differences between these heat sources and VHD appear small, but near the margin VHD exceeds the expected values of   GHF by one to two orders of magnitude. At present, VHD releases more heat than GHF from ~50 km inland to   75 km up the flowline (< 75 km inland due to a sinuous path) to   the margin. In the future, when larger volumes of water flow from farther in the interior, the zone where VHD surpasses GHF may increase its reach to ~100   150   km upstream from the ice margin (Fig. 6c).

**6 Discussion**

**6.1  The impact of VHD**

sumption ~~that subglacial water pressure is equal to ice overburden pressure everywhere. The latter represents a reasonable approximation for distributed subglacial drainage systems (e. g. Engelhardt and Kamb (1997), Fountain (1994), and Meierbachtol et al. (2013)). There are no conduits in this model, although it is likely that conduits would form along the paths where maximum flow and heating occurs (Fig. 12.1 and 3). Conduits are unlikely to form deep in the interior and in the well-studied Russell/Leverett region have been observed up to 34 km inland . When conduits do form, they should draw down local~~ An increase in the supply of surface runoff to the bed will lead to an increase in subglacial VHD as the climate warms in the future. We have shown that a six-fold increase in VHD is predicted by the end of the century under RCP 8.5. The impact of this increase is uncertain. This is because other results show that glaciers can either increase Zwally et al. 2002, (Zwally *et al.*, 2002) or decrease Sundal et al. 2011; Tedstone et al. 2015, (Sunday *et al.*, 2011; Tedstone *et al.*, 2015) their mean annual velocity as additional water accesses the bed. The theory of efficient versus ineffi­cient subglacial drainage explains the different observations, but it is not known what is the current mode of subglacial water drainage beneath all parts of Greenland, what specific thresholds may cause switches in drainage modes, nor the associated response of ice dynamics to these switches.

When discussing subglacial hydrology, a simplification can be made that increased water input should lead to increased basal lubrication and faster sliding but increased VHD  subglacial water pressures  which will result in increased VHD wherever pressurized subglacial water drains from the distributed drainage system into the lower-pressure conduit system. Since the current model does not explicitly represent pressure drops into conduits, this will occur farther inland than represented here. Our model may therefore be overstating the concentration of VHD and slower sliding. However, because VHD is generated by water flow, some condition is needed to define which behavior is dominant in a given setting. We speculate that increasing VHD will have different impacts near the ice sheet margin and development of conduits will spread the heat dissipation more evenly over areas further inland. as compared to the interior of the GIS. Steep hydropotential gradients, found often near the margin, favor high VHD generation for a given water discharge while small gradients do the opposite. At the same time, under thinner ice near the margin, pressure available to close subglacial conduits is smaller than under thicker ice.

**6.2 Heat at the ice sheet bed**

The marginal zone ice response to VHD has been well-studied and observed in the southwest sector, where an increase in runoff is correlated with reduced glacier velocities

Bartholomew et al. 2010; Sundal et al. 2011; Tedstone et al. 2015, (Bartholomew *et al.*, 2010; Sundal 201
The interior ice response to increased runoff is less well-studied. However, Bartholomew
et al. (2011, Bartholomew *et al.* (2011)) show that that ice does not slow down later in the
season as more runoff reaches the bed, and Doyle et al. (2014, Doyle *et al.* (2014)) shows a
610   year-on-year increase in velocity even with increasing runoff.

Flow routing and VHD are similar, but not the same. Basal water is routed based on
the change of the net hydropotential (Eq. **??**, Fig. 2). Viscous heat dissipation (Eq. 2,
Fig. 3) is similar to $V$ times Eq. (**??**), but Eq. (2) has a second term on the RHS, which
states that 1/3 of the heat that could be released due to thinning ice remains in the water,
615   raising its temperature along with the PTT. For certain elevation changes $Q$ is negative
and basal freeze-on may occur (blue pixels in Fig. 3). Locations of basal freeze-on in the
model may be due to the physical processes described above, or due to the basal DEM
not actually representing the basal topography and flow paths. Where basal re-freezing
occurs, additional heat is released and new warm ice is generated   . Predicted locations
620   of basal freezing are near where Bell et al. (2014) shows basal freeze-on packages, but
do not exactly line up, suggesting there may be an error in either this model, the basal
topography from Morlighem et al. (2014) input to the model, or that some advection
occurred between results here and results in Bell et al. (2014)  Mayaud et al. (2014, Mayaud
*et al.* (2014)) has bridged the gap spatially between the margin and the interior, and
625   temporally between present and future, using the same runoff and RCP scenarios used in
this study. They show that near the margin in the Paakitsoq region, conduits are likely to
form earlier, remain longer, and reduce glacier velocity under RCP 4.5 and 8.5 compared
to present. They also hypothesize that under thicker ice, conduits are unlikely to form,
and increased water input into a more distributed subglacial drainage system may lead
630   to an increase, rather than a decrease, in glacier velocity.

An increase in the interior ice velocity and a decrease in marginal velocity suggests that
surface slopes and driving stresses will change, a result confirmed by Shannon et al.
(2013, Shannon *et al.* (2013)). However, the range of possible results is not well enough
constrained there to know the impact of the change in driving stress  .

635   Because the heating term is

**6.2   Increasing VHD**

The contrasting impact of VHD on ice velocity appears to be primarily   a function of water
flux rate, it is spatially heterogeneous at the glacier bed. Flow routing creates subglacial
conduits systems that accumulate several orders of magnitude more water than the
640   surrounding non-conduit region (Fig. 12.1) and along a flowline water flux will increase
(Fig. 6b). Variations in bed topography and ice thickness create variations in   basal
pressure and subglacial hydrological flux (e.g. Schoof (2010, Schoof (2010))). However,
the rate of change of $\phi$ along the flowline (Fig. 6a ), which leads to heterogeneous heating,
and variations in melting rate along the flowline   specific conditions that cause a velocity
645   increase or decrease have not been well defined. Andersen et al. (2011, Andersen *et al.*
(2011)) performed a sensitivity study between glacier velocity and increased runoff, but

that study was limited to 55 days and one marine terminating glacier. It is therefore hard to estimate what the impact of increasing VHD will be in the future on the GIS.

**6.2.1 Increasing VHD in the marginal zone**

A threshold of a 50% increase in runoff has been identified by Tedstone et al. (2015, Tedstone *et al.* (2015)) as leading to a widespread reduction in glacier velocity in the southwest sector marginal zone. It is likely that the threshold is not a 1.5 times increase in runoff, but perhaps the absolute change in VHD. We demonstrate this by comparing both the change [units J ] and relative change [units % ] in VHD in our results to the relative change reported by Tedstone et al. (2015, Tedstone *et al.* (2015)).

We show a 10-20% increase in basin-cumulative VHD over the same region and time period used by Tedstone et al. (2015, Tedstone *et al.* (2015)) (Fig. 6c). At present when using a 5 km grid, the cell with the maximum discharge experiences a flux of 7 km$^3$ 5, right panel). However, our results include all of the GIS and relative decreases elsewhere do not match observed velocity trends over similar time periods (e.g. Rignot and Kanagaratnam (2006) and Joughin et al. (2010, Rignot & Kanagaratnam (2006); Joughin *et al.* (2010))). We also show that this 10-20% increase is equivalent to an absolute increase of ~2 PJ year$^{-1}$ ( per basin (Fig 4a). Over this same period, much of the rest of the ice sheet has near zero or negative increases (when viewed on a petaJoule scale). If the correlation between our results and Tedstone et al. (2015, Tedstone *et al.* (2015)) shown here is causal, then it appears that an increase of VHD on the order of a 1 PJ may be near the threshold that causes a reduction in ice marginal zone velocities.

Under the RCP 4.5 scenario, every basin will experience a 2 % of the annual runoff from the entire ice sheet ). That percentage remains roughly unchanged, and by the end of this century under RCP8.5, one subglacial conduit may discharge up to 32 km$^3$ PJ year$^{-1}$ (Fig. 12.1 increase in VHD, with many gaining >10 PJ year$^{-1}$ (these numbers are the total change of a rate, not a rate of change ). These large water fluxes focused into parts of the model domain mean that up to 10 W m$^{-2}$ can be dissipated in such locations (Fig. 3 and 6c) significant increases in VHD should cause conduits, where they do form, to form more quickly and grow to larger dimensions than they do at present .

We assume that all the heat released from the water is used to melt ice and most of that to form subglacial conduits, and that most conduits form due to VHD from surface runoff because runoff-sourced water dominates basally-produced water, and as shown here VHD dominates GHF. This new basal-source meltwater leaves the ice sheet as latent heat (Fig. **??**). This additional melt represents a small fraction of the total runoff - it is approximately 2%, or 7 km$^3$ year$^{-1}$ at present and 32 km$^3$ year$^{-1}$ under RCP 8.5 near year 2100. Although small by percent, the total volumetric increase is important when one considers that the bulk of subglacial conduit formation is represented by that 7 (present)or 32 (future) km$^3$ year$^{-1}$. The VHD results presented here match the location and magnitude of an increase in runoff that causes a slowdown near the margin according to Tedstone et al. (2015, Tedstone *et al.* (2015)). This supports our proposition that significant increases of VHD around all of GIS in the future may cause a

slowdown in marginal zones elsewhere. At marine terminating glaciers, this effect may be less important than other processes determining glacier velocity and its variability, such as the processes related to ice-ocean interactions (e.g. Walter et al. (2012, Walter *et al.* (2012))). There may be fundamental differences in VHD between marine- and land- terminating glaciers. Relative to land-terminating glaciers, marine-terminating glaciers have a depressed PTT at the discharge location. They may also have reduced surface slopes near their margin and different basal topography (producing different hydropotential gradients), from the cumulative effect of a different flow regime due to their marine boundary.

Regardless of the partitioning of the energy between melt and other processes not addressed here, the relative percentages will likely remain near their present value. Therefore, ~5 times the amount of energy is available for subglacial conduit formation under RCP 8.5 in year 2100.

**6.2.2 Increasing VHD in the interior**

Some future additional heat Future VHD will be distributed over a longer fraction part of a year relative to the present since climate warming prolongs the melt season in Greenland Hanna et al. 2008, (Hanna *et al.*, 2008) . It will also be distributed spatially further inland relative to the presentas the subglacial conduit network expands inland. , and in the interior is less likely to form conduits Dow et al. 2014, (Dow *et al.*, 2014). Additional heat and water at the bed will warm the basal ice. If it cannot be evacuated efficiently by conduits, it will also increase basal water pressures and reduce friction. Given that the primary cause of velocity decreases near the margin is assumed to be the evolution of subglacial conduits reducing basal water pressures, their absence in the interior means we expect velocities to increase, in line with existing observations Doyle et al. 2014; Bartholomew et al. 2011, (Doyle *et al.*, 2014; Bartholomew *et al.* 2011).

When VHD occurs in new locations at the GIS bed it may convert a frozen bed to temperate and increase ice sliding . However, even if some future additional VHD reaches new locations inland and new times of the year due to an increased melt season, the bulk of it will occur in the same place, but it will have higher magnitude. The long-term effects of increased basal water on ice velocity are uncertain, but it appears that short-term velocity increases , especially due to variable input , may lead to overall summer acceleration but annual deceleration , and decadal slowdown . At marine terminating glaciers, the process ought to be similar, but the effect is likely to be less important than other processes determining glacier velocity and its variability (e. g. Walter et al. (2012)). Parizek and Alley 2004; Shannon et al. 2013, (Parizek & Alley, 2004; Shannon, 2013). This is not likely to impact most of Greenland, where the frozen bed is under ice with a surface elevation > 2000 m and therefore remains isolated from surface runoff. However, the northern sector has a frozen bed in regions where, according to our model, VHD increases markedly under the RCP 4.5 and 8.5 scenarios MacGregor et al. 2016, (MacGregor *et al.*, 2016)

The integrated change

K. D. Mankoff

**6.3  Other uses of energy than VHD**

730  Not all of the incoming energy is converted to VHD and used to melt conduits, warm the basal ice, or warm the bed. The primary use of energy other than VHD is change in heat content of the water itself which needs to compensate for the spatially changing PTT. Classic glacier theory (i.e. Röthlisberger (1972, Röthlisberger (1972))) states that when flow is driven by a pressure gradient, 35% of the available VHD is used internally to keep

735  the water at the phase transition temperature (here termed a "loss"), and the remaining amount is dissipated as heat.

Our results show that different sectors may experience large changes in VHD relative to each other due to changes in the PTT. In practice, losses near 35% occur often - whenever elevation change across a grid cell is close to 0, and flow is driven primarily by a pressure

740  gradient (Fig. 2). Negligible losses, near 0%, are also relatively common, when ice thickness does not change and both surface and bed elevation have similar gradients. Gains of 35% may also occur where the surface remains near flat and the bed drops drastically. Finally, in some places an increasing PTT may consume 100% of the available VHD and basal freeze-on occurs. Although here we use the term "freeze" and display

745  locations of freeze-on in blue (Fig. 3), these regions inject excess heat into the subglacial water and basal ice (not tracked in our model), due to the release of latent heat as water freezes Alley et al. 1998; Bell et al. 2014, (Alley *et al.*, 1998; Bell *et al.*, 2014).

There are some disagreements in the location of basal freeze-on between our model and Bell et al. (2014, Bell *et al.* (2014)) observations. The largest area of disagreement occurs in

750  the upper Petermann catchment (bottom right of Fig. 7). In this area, the model does not estimate freeze-on within a few km of the observed basal ice packages. Conversely, in the northwest sector, several observational transects running east-west appear just southward of similar east-west model clusters of freeze-on locations. It seems likely that these agreements may also indicate an artifact in the basal DEM. The basal DEM is built,   in

755  heat per flowline in shown in Fig. 4, with a minimum cutoff removing all small increases. The change between historical and present (P-H in Fig. 4)shows that at present, most of the increase is found in the sector where Tedstone et al. (2015) suggests a 50% increase in runoff has led to a reduction in glacier velocity. Up to $1.8 \times 10^8$ J more heat is released along one flowline, and a similar amount in several nearby flowlines, in this region. That

760  amount may double under RCP 4.5, and increase by an order of magnitude under RCP 8.5. If the correlation between runoff and velocity is controlled by an increase in basal conduit size, quantity, or duration due to VHD, then in the future under RCP 8.5, a widespread reduction in glacier velocity may occur.    part, from these same Bell et al. (2014, Bell *et al.* (2014)) observational transects Morlighem et al. 2014, (Morlighem *et al.*, 2014). The

765  regular vertical spacing and linear horizontal clustering suggests a processing artifact. Finally, our routing model treats over-deepenings and locations of basal freeze-on the same as other regions, which may be an invalid assumption. Hooke (1994, Hooke (1994)) showed that on mountain glaciers, water preferentially routes englacially as it crosses an over-deepened section, rather than subglacially. Englacial routing in the interior of the

770  GIS may not be as likely to occur as in a mountain glacier, but alternate basal paths may be used by the water to avoid locations favorable for freeze-on.

K. D. Mankoff

It seems likely that part of the cause for the disagreement between our model results and Bell et al. (2014, Bell *et al.* (2014)) observations is because the basal DEM does not accurately represent the bed topography, at least at 150 m resolution. If this is the case, it impacts locations of basal freeze-on, and has some impact on the flow routes modeled here, but should not change the basin-scale results. Those results are primarily a function of the surface runoff volume and location, large-scale ice-thickness, and locations of the outlet glaciers. The path the water takes between the source and sink only impacts local VHD distributions, not basin-scale quantities.

**6.3.1 Geothermal Heat Flux**

GHF is spatially and temporally more uniform than VHDwhich is concentrated in subglacial conduits and , at least near the margin where conduits concentrate the flow. GHF is temporally more steady than VHD, which primarily occurs when surface melt is active. Nonetheless, it is worth comparing the magnitude and distribution of the two. Historically the total VHD of 1.4 EJ year$^{-1}$ under the runoff area was similar to the total GHF of 1.1 EJ year$^{-1}$ in over that same area. That is no longer the case , and by the in our calculations for the recent time period, and although GHF flux does not change, the integrated amount does change because the area of integration changes. By the end of this century under RCP 8.5, VHD will contribute ~9 9.8 EJ year$^{-1}$ while GHF only increases to 1.4 EJ year$^{-1}$ due to a slight increase in the runoff area that reaches the bed (when surface runoff is routed to 2000 m elevation before moving to the bed).

VHD and GHF comparisons and relative changes between present and future are most likely to matter in the region > 75 km upstream of the margin and where VHD is active. This is because the change here a) switches which term is dominant and b) is far enough inland that conduits are less likely to form Dow et al. 2014, (Dow *et al.*, 2014), meaning VHD is more likely to be spatially uniform rather than concentrated in smaller regions.

VHD dominates other basal heating terms considered in some glaciological models (for example, \xdef 9991000 \xdef 9991000 Brinkerhoff:2011Sensitivity). Small Models show that the GIS is sensitive to its basal temperature, with small differences in GHF estimates produce drastically producing significantly different GIS growth scenarios , and observed basal temperature measurements do not always agree with assumptions used in existing models . Increased geothermal heat flux Rogozhina et al. 2012, (Rogozhina *et al.*, 2012) Local GHF highs also coincides with onset of fast ice flow Fahnestock et al. 2001, (Fahnestock *et al.*, 200 . The results of our analysis and these GHF studies suggest that if VHD contributes ~6 times as much heat in the future as historically, it may generally changes from 1 to 2 orders of magnitude less than GHF, to 1 to 2 orders magnitudes more than GHF, it will likely decrease the importance of GHF in modulating spatial dynamics of the ice sheetbecause it will swamp the basal supply of heat from GHF , at least underneath the ablation zone within the zone of active basal hydrology dominated by surface water penetration to the bed .

**6.4  Erosion and sediment transport**

Large amounts of eroded material are also flushed out from under the GIS each year Cowton et al. 2012, (Cowton *et al.*, 2012) . The erosion rates implied by the sediment flux is are already several orders of magnitude above the background ($>$ $>$ 1000 year) erosion rates . Koppes and Marchant 2009, (Koppes & Marchant, 2009). Larger VHD leads to larger conduits, faster water flow velocity, and higher erosion rates and sediment transport capacity. Conversely, slow subglacial water flow does not have as much impact on erosion and sediment transport Hodson et al. 2016; Gimbert et al. 2016, (Hodson *et al.*, 2016; Gimbert *et al.*, If 5 times the amount of water flows along the GIS bed by the end of the century, it may will likely increase sediment removal Bogen and Bønsnes 2003, (Bogen & Bonsnes, 2003) . If increased VHD and water at the bed of the GIS simultaneously causes cause sediment removal rates to increase5-fold for ~100 – 1000 years , while at the same time reducing the glacier velocity Tedstone et al. 2015, (Tedstone *et al.*, 2015) and therefore the production of sediment Herman et al. 2015, (Herman *et al.*, 2015) , the state of the bed may rapidly change over the coming century centuries to millenia from potentially deformable subglacial sediments to rigid bedrock Weertman 1964; Kamb 1970; Tulaczyk et al. 2000; Bouga

**6.5  Model domain The impact of model spatial resolution on results**

The model domain has resolution is a 5x5 km grid for most of the analysis presented here, which means results are smoothed over that area. In reality, subglacial discharges occur approximately on the order of one every 5 km along the coast . For example, a discharge of 1000 $m^3$ $m^{-2}$ $year^{-1}$ of water in a 25 $km^2$ grid cell in Fig. 12.1(8.5) equals 25 $km^3$. That same 25 $km^3$ might discharge through a conduit on order $10 - 100$ m wide, rather than the 5000 m wide cell used in the analysis. Lewis and Smith 2009, (Lewis & Smith, 2009). If a single conduit on the order of $10 - 100$ m wide carries all of the water (Fig. 12.1) and is subject to all of the heating (Fig. 3), then values reported (here spread over 5000 m) are likely be one or more orders of magnitude larger in small focused regions, and one or more magnitudes orders of magnitude smaller outside the conduit. This limitation of the model domain is less important in the interior, where conduits are less likely to form.

Our treatment of en- and sub- glacial hydrology is simplified because it does not represent actual conduits, but is at the same time more comprehensive than in existing global climate or ice-sheet models (e.g. Pollard and DeConto (2012, Pollard & DeConto (2012))), and may offer a computationally efficient yet improved method to incorporate parameterizations of VHD at their existing grid resolution.

**7  Conclusion**

Large volumes of supraglacial runoff observed in Greenland ablation zone, including at relatively high elevations above sea level, contain large amounts of gravitational potential energy . We estimate that approximately 65% of this energy is The high potential energy contained in large volumes of GIS surface meltwater is largely dissipated as heat at

K. D. Mankoff                                                                          p. 22 of 33

the ice sheet bed in the ablation zone. This . This dissipated energy averaged 1.4 EJ year$^{-1}$ over the 1900s between 1960 and 1999 , but has recently increased to 1.9 2.1 EJ year$^{-1}$ and will likely increase to 3.2 or 9.0 3.5 or 9.8 EJ year$^{-1}$ by the end of the century under RCP 4.5 or 8.5 respectively. Viscous heat dissipation will become This viscous heat dissipation by subglacial water is the dominant basal heat source , and under near the margin, and its impact will move inland due to increasing flux, even if conduits do not form in the interior. Under RCP 8.5 will swamp , VHD will be about ten times larger than the ~1 EJ year$^{-1}$ contributed by geothermal heat flux in to the same area.

This up Up to 6 times additional future heat VHD at the ice sheet bed (relative to the historical amount) should result in a similar 6 fold increase in basal ice melt volume and may is expected to contribute to more numerous , larger, longer-lasting, and more widespread subglacial conduits . The effect of increased conduit formation is not captured by this model, but based on recent results and subglacial theory , in the margin zone. Based on recent measurements by others and glaciological theory of ice sliding, increased VHD may decrease glacier velocity at the margin, and accelerate it in the interior where conduits either do not form or have insufficient impact on subglacial water pressures to influence ice sliding rates . This decrease may be offset by other processes and there may still be a net acceleration, especially at marine terminating glaciers. Along with possible impacts on ice velocity due to changing subglacial conduit configuration, increased runoff will remove more sediments, which will likely change the stress state at the glacier bed by changing the glacier/till/bedrock interface.

**8    About This Document**

This is an attempt manuscript is prepared with the intent to create a "fully reproducible" scientific publication. We may not have completely succeeded, but have made progress in this direction. In order to be fully reproducible at the binary-level, a clone of our operating system with the full analysis software should be provided. This could be done with a virtual machine (VM) but we have not taken this step because VMs require ~20 GB of space, and journals do not yet support this type of supplemental material size of supplemental materials .

Instead, we used only free and open source software above the operating system level, document in detail the version(s) of all software packages used, and provide every line of code required to reproduce the document, beginning with the commands to download the MAR Fettweis et al. 2013, (Fettweis *et al.*, 2013) and IceBridge BedMachine Greenland, Version 2 Morlighem et al. 2015; Morlighem et al. 2014, (Morlighem 2014, 2015) data sets, followed by the grass GIS GRASS GIS Neteler et al. 2012, (Neteler *et al.*, 2012) and Python commands to produce intermediary data products and graphics.

The supplementary data is a plain-text file that contains the manuscript text and all of the code. As plain text, it can be viewed in any editor or document viewer. However, it's its internal structure is that of an Emacs Org Mode Schulte and Davison 2011; Schulte et al. 2012, (Schulte file and is best viewed in Emacs, which supports execution of the embedded code blocks.

K. D. Mankoff

A reader should be able to reproduce the contents of this document, although it will require 3rd-party applications (GRASS, Python, etc.), and, optionally, a similar system-level Emacs configuration as the authors.

**9 Acknowledgements Acknowledgments**

We thank D. van As for initial discussions on this topic, D. Pollard for comments, and M. Morlighem and X. Fettweis for providing accessible and documented data. , R. Bell for sharing data, and D. Pollard, Anonymous referees, and The Cryosphere Discussion reviewers for comments. K. D. Mankoff was funded by NASA Headquarters under the NASA Earth and Space Science Fellowship Program (Grant NNX10AN83H) and the Postdoctoral Scholar Program at the Woods Hole Oceanographic Institution, with funding provided by the Ocean and Climate Change Institute. S. Tulaczyk was funded by NASA grant NNX11AH61G. Anonymous referees contributed greatly to the focus and scope of the manuscript.

**10 References**

**11  Tables**

Table 1: Properties of Greenland runoff and viscous heat dissipation. (H)istorical period covers 1900    1960    – 1999, (P)resent spans 2010 – 2019, and the RCP(4.5) and (8.5) periods span 2090 – 2099. Runoff volume and area    from MAR , elevation from MAR combined with IceBridge BedMachine Greenland, Version 2 Fettweis et al. 2013, (Fettweis *et al.*, 2013) . Geothermal heat flux from Shapiro and Ritzwoller (2004)   Shapiro and Ritzwoller (2004, Shapiro & Ritzwoller (2004)) calculated only under runoff area.

| Property |
|---|
| Runoff volume |
| Runoff area   $10^6$ km$^2$   0.68   0.68   0.84   1.23   Runoff mean elevation $\bar{z}_s$   m   1293   1296   1344   139 |
| Potential energy   $10^{18}$ J year$^{-1}$   2.1   2.9   4.9   14.3   Viscous heat dissipation |
| Geothermal heat flux |

**12 Figures**

**12.1 Figure: Schematic Flow-routed Accumulation**

[Figure]

Figure: Gravitational Potential Energy  Gravitational potential energy from surface runoff.  Labels represent (H)istorical mean runoff  from 1900  1960  – 1999, (P)resent mean runoff  from 2010 – 2019, (4.5) years 2090 – 2099 under IPCC AR5 RCP4.5, and (8.5) same as (4.5) but under scenario RCP8.5. Gray contour marks 0 m elevation.
Figure: Flow-routed Accumulation  Accumulation of subglacial water flowing through each grid cell. Results are presented on a 5x5 km grid. Labels same as Fig. **??**. Gray contours mark 0 and 2000 m elevation.

Figure: Gravitational Potential Energy  Gravitational potential energy from surface runoff. Labels represent (H)istorical mean runoff  from 1900  1960  – 1999, (P)resent mean runoff  from 2010 – 2019, (4.5) years 2090 – 2099 under IPCC AR5 RCP4.5, and (8.5) same as (4.5) but under scenario RCP8.5. Gray contour marks 0 m elevation.

Figure: Flow-routed Accumulation  Accumulation of subglacial water flowing through each grid cell. Results are presented on a 5x5 km grid. Labels same as Fig. **??**. Gray contours mark 0 and 2000 m elevation.

Figure 1: Schematic  Accumulation  of a water-filled moulin and a land- and marine-

910   12.2   Figure: $\Delta\phi$ $\nabla\phi$ , $\Delta\phi_z$ $\nabla\phi_z$ and $\Delta\phi_p$ $\nabla\phi_p$

[Figure]

Figure 2: Net hydropotential change between gradient for each cell and its downstream neighbor ($\Delta\phi$ $\nabla\phi$) , and the decomposition of net hydropotential change gradient to elevation-driven hydropotential gradient ($\Delta\phi_z$ $\nabla\phi_z$ ) and pressure-driven hydropotential gradient ($\Delta\phi_p$ $\nabla\phi_p$ ). Red is positive and blue is negative. Negative $\Delta\phi_z$ implies flow uphill driven by a pressure gradient, downhill and positive $\Delta\phi_p$ implies flow driven by thinning under thining ice. Large differences in released heat (~66%) are due to Blue implies flow uphill and under thinning or thickening ice.

12.3   Figure: Heat

[Figure]

Figure 3: Heat released at the bed due to VHD . Labels same as Fig. ?? 12.1 . Colorbars represent heating (red) and cooling (blue), and numbers are valid for both colorbars (i.e. 1 $W\,m^{-2}$ equals $3x10^7$ J $m^2$ $year^{-1}$, and each of those values are positive on the red colorbar and negative on the blue colorbar). Gray contours mark 0 and 2000 m elevation.

**12.4 Figure: Absolute Δ heat per basin**

[Figure]

**Figure 4:** Change in VHD per basin. Label TB-TR represents increase from reference to baseline periods (1985–1994 and 2007–2014) from Tedstone et al. (2015, Tedstone *et al.* (2015)), P-H increase from historical to present, 4.5-P increase from present to 2090s under RCP 4.5, and similarly for 8.5-P. Gray contours are 0 and 2000 m elevation.

**12.5   Figure: Relative Δ heat per basin: Tedstone (2015) comparison**

[Figure]

Figure 5: Relative change in VHD per basin highlighting the impact of different averaging periods. P/H is 100 x present divided by historical, and TR/TB is 100 x the reference (1985–1994) divided by baseline (2007–2014) years from Tedstone et al. (2015, Tedstone *et al.* (2015)). Gray contours are 0 and 2000 m elevation.

12.6    Figure:  Flowline

[Figure]

Figure 6: Detail along a flowline on Russell glacier  in Southwest Greenland  . a) surface and bed elevation (left axis) and change in   gradient of   $\phi$ (right axis), b) flow rate of subglacial water  , and c) power or heat flux  , frictional heating, and geothermal heat flux (GHF)  . Legend labels H, P, 4.5, and 8.5 same as Fig. **??**  12.1  . Frictional heating from Brinkerhoff et al. (2011)  Brinkerhoff et al. (2011, Brinkerhoff *et al.* (2011)) , and geothermal heat flux (  GHF )   from Shapiro and Ritzwoller (2004)  Shapiro and Ritzwoller (2004, Shapiro & Ritzwoller (2004))  . Lines in panel (c) are smoothed to reduce visual noise and are actually as variable as panel (a).

12.7    Figure:  Δ Heat   Bell 2014 comparison

[Figure]

Figure 7: Change in cumulative upstream heat released  Close-up of Petermann glacier region  at the bed  150 m resolution  . Label P-H represents increase from historical to present, 4.5-P difference between present and 2090s under RCP 4.5, and similarly for 8.5-P Gray basemap is shaded relief of hydropotential gradient  . Symbols show amount  Blue dots and black lines are locations  of heat released between terminus  basal freeze-on predicted by the model  and upstream source for each flowline  from Bell et al. (2014, Bell *et al.* (2014)), respectively  . Numbers (units are Joule) are  Each blue dot is  a legend for the changing size of one symbol  150 m x 150 m square pixel  .Increases of less than $2x10^8$ J are not shown.

13  TC

---

## Author Response (AR2)

**Reply to Reviewers**

Ken Mankoff & Slawek Tulaczyk

**Comments from reviewers are in normal font and differentiated from the replies that use a bold colored font.**

**Contents**

**1   Editor**

I have had now three new reviews from the previous reviewers on your manuscript. They all three acknowledge on the fact that the second version was greatly improved from the first one. Nevertheless, their reviews, even if being very contrasted, still indicate that your paper can and must be improved before it reaches the standard for being published in TC. I will then take the final decision on the basis of your reply to these new reviews and a third version of the paper. Importantly, this new version should clearly highlight that you are not just re-expresssing the simulated increase in runoff in modified units and clearly state the limit of the approach about the effect of potential for subglacial channelisation. This will require rewording of some parts of the manuscript.

**We are grateful for the editor's support of this manuscript, and try to be sensitive to the disparate reviews from this last round. We address the two specific points you have raised (not just modified units, limits of approach) in the revised version.**

**2 Referee 1**

The revised paper is much improved from the original. I still find the whole calculation rather simplistic, and am not convinced that it provides new understanding, but I can accept that framing discussion in a different language may be helpful to some readers. The presentation of the revised manuscript is more coherent than the original, with more focus on results (figures), and I think this now makes it a helpful contribution, despite my reservations mentioned above and in the previous review.

I would now recommend publication, but I have some minor comments that should be considered.

**We are pleased to hear the referee recommends publication pending consideration of the comments. We agree that the calculations are simplistic, but point out that this is not a numerical modeling paper, but rather a discussion paper on changing hydrology. The focus is not on the calculations, and we view their simplicity as a positive feature of the paper. Adding more complicated equations may provide additional insights, but given that the system is highly underconstrained, many terms in any additional equations would be untested parameterizations, and we feel that would detract from this paper.**

In equation (1), make clear the partitioning into phi_z and phi_p - I don't think these are explicitly defined anywhere.

**Done.**

It may be preferable to describe the viscous heat dissipation in Watts, rather than J/year, as that would remove some of the ambiguity about increases over time (as rather confusingly raised in section 4.4). I doubt that units of PJ/year are any more informative to most readers than GW or MW.

**We originally wrote the manuscript with Watts, but switched to Joules because the energy is not released uniformly in time. The 2.1 EJ year$^{-1}$ can more easily be conceptualized as 2.1 EJ per summer and 0 EJ per winter, while the equivalent 67 GW implicitly suggests a steady state output.**

Last paragraph of section 4.4, the sentences don't seem to agree with each other. One says that your results match the results from Tedstone et al; and then the last says there is a difference (10-20% increase rather than 50% increase). This might make sense if you mean that the runoff increase matches Tedstone et al, but the VHD increase is different; but then the explanation for the discrepancy does not really make sense, since this explanation seems to imply that the simulated runoff is incorrect. In any case, I am slightly confused why you treat the different increases in runoff and VHD as a discrepancy that needs explaining - there is no reason why they should be the same, since the VHD depends on the potential (effectively elevation) of the runoff as well as the magnitude.

**We meant to state that they roughly (increase in that sector), but not exactly (10-20% v. 50%) match. We have clarified the wording. We agree with the reviewer that an agreement between the Tedstone runoff increase and our VHD increase is not**

**necessary, as they are different things. However, they are closely related, based on current subglacial hydrological theory. We think it is instructive to point out reasons for the possible discrepancy, and now include the point raised by the reviewer too.**

The flowline in figure 6 is presumably an ice flowline, since you seem to have regions of negative potential gradient which suggest it is not possible for the water to be flowing along this line. Would it make more sense to follow a path taken by the water (ie. one of the flow paths obtained from the routing algorithm)?

**The flowline is a hydrological path following the maximum hydropotential from source to outflow, not an ice-flowline. There are not regions of negative potential gradient, but are regions of near-zero potential gradient. These occur if the gradient across a grid cell is near zero, and are likely indicative of small subglacial routes below the model resolution. The upstream and downstream grid cells probably have lower gradients, with the losses there contributing the gains in the grid cell between. Clarifying text added in section 4.5.**

In a couple of places (line 2, p8; line 5, p 11) it is suggested that viscous heat dissipation can have the effect of both accelerating and decelerating flow. Is this really what is meant? I think that is true when talking about increased runoff, because runoff has the effect of increasing pressures as well as increasing VHD (which then opens drainage space and lowers pressures). But do you have in mind a mechanism by which VHD itself causes an increase in ice velocity (rather than the runoff that is causing the VHD)?

**We meant increased runoff and have adjusted the text. We could not find an issue with line 5, p. 11, but checked our usage of VHD v. runoff elsewhere in the text.**

In section 5.2.1 (line 9, p8) it is suggested that it is 'likely' that there is a threshold change in VHD that leads to ice velocity decrease. This seems speculative to me, and I think should be phrased 'We suggest', or 'We speculate', rather than 'It is likely'. It seems well established that the effect of VHD might be to facilitate a decrease in velocity, but I do not see the evidence here for a threshold. It is also not clear what the 'baseline' for this change is (i.e. when should I measure the change from?).

**Yes this is highly speculative and we have changed the wording to reflect that. We do not have specific advice about a "baseline". Here we are simply using the same baseline as Tedstone et al. (2015).**

**3    Referee 2**

Aspects of this work have been greatly improved since the initial draft, notably a much simplified Methods section, and the idea behind the work remains interesting. However, other aspects of the manuscript still leave much to be desired, including the organization of the paper (which is muddled in places), the extensive comparison to Tedstone (which was unconvincing), and the majority of the Discussion text (which almost wholly lacks

synthesis). Unfortunately I do not think that the manuscript meets the standard of the journal at this time.

**We are glad to read that the reviewer considers the manuscript greatly improved and interesting. We have revised the latest version using your suggestions in the hope that it meets the standard for publication.**

3.1  Improved Methods

The work that the authors put into this section shows well. The methods are much more condensed and clear. However, I do still find some issues.

- I remain convinced that the outflow elevation of marine glaciers would be correctly defined as sea level, as AR1 also pointed out. The authors' treatment of this (P4 L10-13, P8 L27, and responses to reviewers) states otherwise. In my understanding, a parcel of water discharged at the base of a glacier may remain at that elevation, but it displaces a parcel of water upward, which itself displaces the next parcel upward. When these gains in potential energy are integrated, the net result is sea-level potential. However, this is only a small worry at this point.

**We agree, but the parcels described above are no longer underneath ice, and therefore outside the domain of the study, and the scope of this manuscript. These waters parcels rising vertically up marine terminating ice face are similar to the initial water falling down a moulin. The model begins at the moulin bottom with a pressure term (a function of the ice thickness), a depressed PTT, and the only elevation from the bed elevation. We no longer consider the water at the surface, falling down the moulin, etc. in this version, just as we do not consider the PTT change you describe above. Furthermore, one can argue the temperature of the water falling down the moulin may matter (it melts open the moulin), but the thermal issues raised above at the marine edge are insignificant compared to the heat content of the fjord waters in contact with the ice.**

- I am confused that the authors study only basal potential energy (rho*g*b) while discarding potential energy from the surface (rho*g*s), which is much greater in magnitude. The paragraph on supraglacial rivers suggests that the authors recognize this (P3 L16-17 especially). Perhaps my memory of the first manuscript colors this, though.

**All versions of this manuscript have only focused on energy at the bed. The original manuscript did consider the total potential energy, but only as a conceptual overview, before arguing that all that energy was released at the bed. However, the current manuscript does include $\rho g s$ in the RHS of Eq. (1).**

- The treatment of PTT in the calculation seems to be inconsistent (P5 L11-14). The calculation ignores PTT changes when ice thins along the basal hydraulic gradient, but includes it when ice thickens along the water path.

**The calculation does not *ignore* PTT under thinning ice. Under thinning ice, VHD is**

**reduced by the PTT, and we state that it is "not included in our VHD numbers". It is included in the calculation, but subtracted from the VHD value, hence the poorly-worded "not included". We have rephrased this to be clearer.**

3.2   Organization

The manuscript suffers from poor organization. For instance:

- In Results, a methods paragraph (P4 L26-31) appears.

**Fixed.**

- In Discussion, a results sentence appears for the first and only time in the paper (P7 L8).

**Fixed.**

- The Discussion is repetitive of the Introduction (P7 L21-25 especially, with 4 of 5 references repeated for the same facts).

**We do not think that the occasional repetition is inappropriate.**

- There may be others. These basic offenses contributed to the difficulties I had in following the logic of this work.

3.3   Tedstone

In the revision, the authors added a substantial component to their analysis by attempting to parallel the Tedstone et al. (2015) study. To their original Historical / Present comparison (which I thought was good), the authors added a Reference / Baseline pair as well (which I found hard to keep track of, and which I think added little), based on Tedstone. I could not find that any of the reviewers asked for this addition.

**Tedstone was not added at the specific request from a reviewer, but because we think it connects the modeling work presented here to an independent observational result. We have clarified this in the text, at the end of the Introduction section. We have clarified the reference and baseline pairs when we introduce them in the Data section, and mention them later in the ethods section.**

The authors bluntly state that one of the Tedstone conclusions is "likely" not true (P8 L9-10) and provide an alternative hypothesis.

**We have rephrased this to be more speculative.**

I found the support for this new idea (section 5.2.1) entirely unconvincing and difficult to even understand the authors' path of reasoning. The authors say that if the increase in VHD in the Tedstone basin caused the observed slowdown there (P8 L12), then this is incompatible with the ice-sheet-wide decrease in VHD observed over that period having caused the generally observed ice-sheet-wide speedup (P8 L14) — although it seems to me that these ideas do not contradict one another at all. (I am putting aside obvious issues with comparing velocity changes observed over slightly different time periods, and

extrapolating land-terminating ice in western Greenland to the whole ice sheet.) But a few paragraphs later, the authors contradict themselves by extrapolating the Tedstone observations over the whole ice sheet after all (P8 L23-24). Such inconsistencies greatly undermined my confidence in this work.

**We have rewritten parts of Section 5.2.1. due to the reviewers comments. The idea that a future increase in VHD elsewhere in Greenland may lead to a slowdown near the marginal zone is not a contradiction to the earlier text in this paragraph. We now speculate that at present, an increase in VHD is correlated with a decrease in velocity. We point out elsewhere there is presently a decrease in VHD relative to the past. We do not compare with velocity elsewhere, because we now feel that the velocity change maps in Rignot et al. (2006) and Joughin et al. (2010) are not appropriate. They are not GIS-scale, or even regional-scale. Rather, they focus only on the outlet glaciers, which is a different scale than the work reported here.**

3.4   Discussion

The immaturity of the Discussion and the failure of this study to integrate its results with past work are major issues. While the authors include relevant citations, they do so largely in the form of a literature review. They do not demonstrate how their results build on this literature, except in the broadest strokes (i.e., VHD enhances conduit formation, which slows ice near the margin but not far from the margin) which were already made clear in the introduction. In short, the authors leave unclear what their work adds to the community's understanding.

Upon a closer read, I did find a few sentences where the authors introduce a nice (and perhaps novel) line of reasoning / speculation about the competing effects of increasing runoff and increasing VHD (P7 L14-17). However, that idea is buried deep within the text and not revisited. The rest of the subsection (3 paragraphs), which would normally be used to build on the idea, consists only of a rambling literature review. While I remain interested in this topic and would enjoy continuation along this line of thought, I am at a loss for specific suggestions at this point.

**The competing runoff and VHD discussion is expanded in sections 5.2.1 and 5.2.2.**

Alternately, it may still be possible to frame the paper differently from runoff / speedup — for example, around GHF vs VHD, which is presently an interesting section, and one that perhaps could be expanded toward greater ideas. (In fact, the abstract suggests that this is the main point of the paper, but this is not borne out.) But as it is framed now around ice sheet mass balance, the paper makes only a vague contribution. At this point, I agree with AR1 who said that the study merely converts future runoff rates into future VHD rates, which was not particularly illuminating.

**We were strongly encouraged by previous reviews to not compare to GHF, and we have reduced that section. Although we mention it in the abstract, we do not consider it a primary result. We respectfully disagree with this reviewer, and their interpre-**

**tation of AR1's review, that a conversion of "future runoff rates into future VHD rates" is not useful. The flow routing, tracking of changes in PTT, GIS-wide perspective, consideration of spatial variability around the GIS, and consideration of future changes, combine to move this work beyond simple units conversion as suggested by the reviewer.**

Some of the section headings (4.1 Volume of subglacial hydrology; 4.3 Flow-routed spatial distribution of VHD) are inscrutable.

**We have improved the former. We are not sure what is inscrutable about the latter.**

**3.5 Figures**

Figure 2 caption — how could water flow both uphill AND toward ice that thickens along flow?

**It can't. The statement is referring to different panels of the figure. We have changed "and" to "or".**

Figures 4/5 — Tedstone basin should be indicated

**Done.**

Figure 7 — it is hard to compare the small blue dots with the large black bars. Lightening the background relief map might help, as well as plotting the dots on the top layer and using colors that provide better contrast than blue/black.

**We have enlarged the blue dots and provided better contrast.**

**4 References**

[revised manuscript text omitted]

---

## Author Response (AR3)

**Reply to Editor**

Ken Mankoff and Slawek Tulaczyk

**Comments from reviewers are in normal font and differentiated from the replies that use a bold colored font.**

Dear Kenneth,

I have now studied this new version of your paper based on this second round of reviews. I have still some issues with some points that need to be corrected or clarified before your paper can be accepted in TC.

**Dear Olivier,**

**Thank you for your continued support of this manuscript. I am grateful for the time and consideration you have put into it. The latest revisions address all of your comments.**

My first issue is regarding two important assumptions on your approach that are not enough discussed in the current version of the paper. These two assumptions are that:

1] the area concerned by runoff input is constant over the period of simulation. At the front, it is bounded by the current geometry (see my second point) and upstream the limitation is based on the 2000 m contour level which is considered as the maximal altitude for crevasses opening from a work by Poinar et al. (2015). For me, it looks highly speculative that this elevation will not change in the future especially if more melt is expected at higher elevations.

**We added the 2000 m limit due to earlier comments from reviewers, but are happy to discuss its implications and limitations in more detail. We now show results for the RCP 8.5 scenario without the 2000 m limit in Table 1 and the area where that runoff occurs in Fig. 2. We discuss the 2000 m limit at 6 different locations in the revised text. Note that the only significant change is the total geothermal heat flux which increases due to the increased area of integration. The values discussed in the paper, "integrated VHD" and "maximum VHD" are not changed.**

The fact that the area at the base is constant certainly induce higher value of energy per m2 than if it was allowed to evolve with time. This assumption should be discussed and high values for future presented in W/m2 should be discussed more carefully.

K. D. Mankoff

We disagree with the above issue. We think that the values reported here are a lower bound than if the 2000 m elevation limit were removed and the margin were allowed to retreat inland.

We only discuss the maximum energy per m$^2$, and the integrated energy. The maxima, which occur at the margin discharge locations, are not changed by moving the initial location. The integrated amount is also unchanged in this case - the area increases, but the total energy only increases by a few percent.

In fact, because we limit the surface waters to <2000 m, in some locations we decrease the total energy, by decreasing the initial elevation. This is not significant though, because only small volumes of surface melt occur at elevations >2000 m.

If the ice surface evolved, the near-margin gradients would likely be steeper, and the energy released would increase. The increased gradients would likely offset the small (order 1 pixel in this model) decrease in area due to margin retreat. We do discuss the potential impact of changing surface gradients, and now explicitly point out that this is not part of our model.

2] the upper surface elevation is not updated in the future, and from your equation (1) it is obvious that the gradient of the hydropotential is much more sensitive to the surface slopes than the bed slopes.

Correct, the hydropotential is ~9x more sensitive to surface slopes than to bed slopes. But it is also common that bed slopes are greater than surface slopes (although perhaps not 9x greater), so the net effect of the surface and bed on flow routing is less than 9:1.

We now explicitly point out that the surface is static. We also briefly discuss where and why the bed gradient dominates the surface gradient (in the interior, where the surface gradient is smallest).

If you melt the glacier fronts, upstream slopes will increase, changing the routes of water at the glacier base. I agree that accounting for change in surface elevation would require an ice flow model which is certainly out of the scope of your paper, but anyway you should mention and discuss this strong assumption in your model results. As far as I can see, the gradient of the hydropotential is constant in your prognostic simulation? This should be mentioned. Also, as a consequence of a fixed surface elevation, the retreat of the margins is not accounted for (see my first point).

The above issues are addressed in the revised manuscript.

One other important point is that, as mentioned by referee #1, it would be less confusing to express the heat dissipation in W rather than in J/a. The fact that the energy is not released uniformly in time is not an argument as anyway the melt seasonality is not discussed in your paper.

Units have been changed throughout the manuscript and figures.

I have some other minor points that should also be corrected (page number refer to the

K. D. Mankoff

coloured version of the paper):

- everywhere in the manuscript, "year" for units should be "y" or preferentially "a"

**Fixed. We now use a.**

- abstract, line 9 and 10: I would suggest to write in EJ per year (or each year).

**We now use W (or GW) and Joule is no longer in the document.**

- page 2, line 29: is it an ice flow line or a water flow line as stated in the response to referee #2?

**Now phrased as "subglacial water flow line".**

- page 4, line 15: changing phase change temperature?

**Rewritten as "changing PTT".**

- page 5, line 29: discuss the hypothesis of the 2000m limit

**Done, and we discuss it in other locations in the document also.**

- page 6, line 6: the wording thinning or thickening is a bit confusing as you are not evolving the surface elevation. I would suggest to use "for ice thickness decreasing downstream, ..." instead?

**Done.**

- page 7, line 4: W would be better and would avoid the sentence in bracket which is confusing.

**Done.**

- page 7, line 20: not a reanalysis?

**Reworded.**

- page 7, line 24: specify what is a flow line (ice or water)?

**Done.**

- page 7, line 25: increase in water volume? not flow?

**Done.**

- page 10, line 19: problem in the sentence?

**Fixed.**

- page 10, line 21: again, would be clearer if in W

**Done.**

- end page 10: the case of land-terminated glaciers is not discussed here

**Clarified.**

- page 12, line 3: east-ouest vs south-north more than horizontal vs vertical?

**Fixed.**

- page 13, line 20: you approach would not be useful for an ice flow model which need the water pressure or effective presssure (and which is fixed in your approach).

**Clarified.**

[revised manuscript text omitted]